# Activation mechanism of ATP-sensitive K$^+$ channels explored with real-time nucleotide binding

Michael Puljung*, Natascia Vedovato, Samuel Usher, Frances Ashcroft*

Department of Physiology, Anatomy and Genetics, University of Oxford, Oxford, United Kingdom

**Abstract** The response of ATP-sensitive K$^+$ channels (K$_{ATP}$) to cellular metabolism is coordinated by three classes of nucleotide binding site (NBS). We used a novel approach involving labeling of intact channels in a native, membrane environment with a non-canonical fluorescent amino acid and measurement (using FRET with fluorescent nucleotides) of steady-state and time-resolved nucleotide binding to dissect the role of NBS2 of the accessory SUR1 subunit of K$_{ATP}$ in channel gating. Binding to NBS2 was Mg$^{2+}$-independent, but Mg$^{2+}$ was required to trigger a conformational change in SUR1. Mutation of a lysine (K1384A) in NBS2 that coordinates bound nucleotides increased the *EC$_{50}$* for trinitrophenyl-ADP binding to NBS2, but only in the presence of Mg$^{2+}$, indicating that this mutation disrupts the ligand-induced conformational change. Comparison of nucleotide-binding with ionic currents suggests a model in which each nucleotide binding event to NBS2 of SUR1 is independent and promotes K$_{ATP}$ activation by the same amount.
DOI: https://doi.org/10.7554/eLife.41103.001

*For correspondence:
michael.puljung@dpag.ox.ac.uk
(MP);
frances.ashcroft@dpag.ox.ac.uk
(FA)

Competing interests: The authors declare that no competing interests exist.

## Introduction

ATP-sensitive K$^+$ channel (K$_{ATP}$) closure initiates the electrical response of pancreatic β-cells to metabolic changes induced by increased extracellular glucose (*Ashcroft and Rorsman, 2013*; *Quan et al., 2011*). K$_{ATP}$'s metabolic sensitivity is accomplished through the coordinated activity of three classes of intracellular adenine nucleotide binding site (NBS), one inhibitory and two stimulatory. Despite the recent publication of several cryo-EM structures of K$_{ATP}$ showing these NBSs at near atomic resolution, the detailed mechanism by which energetic contributions from nucleotide binding to each site sum to affect channel gating remains obscure (*Lee et al., 2017*; *Martin et al., 2017a*; *Puljung, 2018*; *Wu et al., 2018*).

K$_{ATP}$ is formed by four inward-rectifier K$^+$ channel subunits (Kir6.2 in β-cells), each associated with a modulatory sulfonylurea receptor (SUR1 in β-cells, *Figure 1a*) (*Inagaki et al., 1995*; *Sakura et al., 1995*; *Aguilar-Bryan et al., 1995*; *Inagaki et al., 1997*). The inhibitory NBS of K$_{ATP}$ is located on Kir6.2 and clearly resolved in several of the cryo-EM structures (*Lee et al., 2017*; *Martin et al., 2017a*; *Puljung, 2018*; *Wu et al., 2018*; *Tucker et al., 1997*). Like other ATP-binding cassette (ABC) proteins, SUR1 has two cytoplasmic nucleotide-binding domains (NBDs), which associate to form two NBSs at the dimer interface (*Figure 1a,c*) (*Lee et al., 2017*). NBS2, formed primarily by the nucleotide-binding Walker$_A$ and Walker$_B$ motifs of NBD2 and the ABC signature sequence of NBD1, is a consensus binding site, based on conservation of key catalytic residues with other ABC family members, and is competent to hydrolyze nucleotides (*de Wet et al., 2007*; *Matsuo et al., 1999*). In NBS1, formed by the Walker motifs of NBD1 and the signature sequence of NBD2, the catalytic Walker$_B$ glutamate is replaced with an aspartate (D854), rendering this site 'degenerate,' that is, incapable of nucleotide hydrolysis. NBS1 binds nucleotides in the absence of Mg$^{2+}$, whereas it is thought that Mg$^{2+}$ is required for binding to NBS2 (*Matsuo et al., 1999*). The NBSs of SUR1

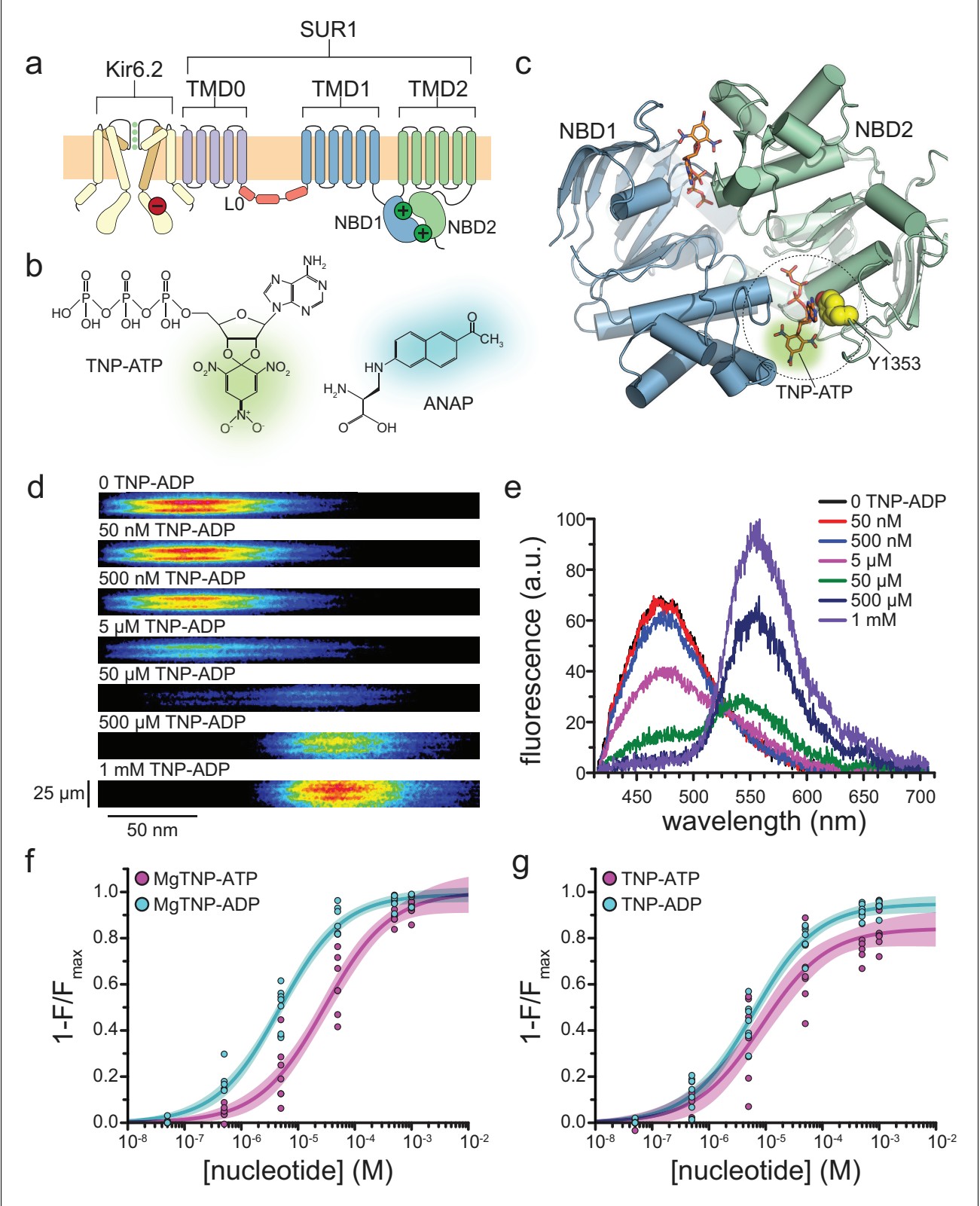

**Figure 1.** Measuring binding to NBS2 of SUR1. (**a**) Cartoon illustrating the topology of the K$_{ATP}$ channel complex. Two (of four) Kir6.2 subunits and one (of four) SUR1 subunits are shown for clarity. The locations of the inhibitory (red, on Kir6.2) and stimulatory (green, on SUR1) NBSs are indicated. (**b**) Chemical structures of TNP-ATP and ANAP. The fluorescent moiety is colored in each. (**c**) Structure of the ligand-bound NBDs of SUR1 (from PDB accession # 6C3O) (*Lee et al., 2017*). TNP-ATP (from PDB accession # 3AR7) was positioned in both NBSs of SUR1 by alignment with MgATP (at NBS1)

*Figure 1 continued on next page*

*Figure 1 continued*

and MgADP (at NBS2), which were present in the original structure (*Toyoshima et al., 2011*). Y1353 from the A-loop of NBS2 is highlighted in yellow to mark the location of ANAP in our experiments. Alignment of TNP-ATP with nucleotides bound to NBS1 and NBS2 in the cryo-EM structure of SUR1 does not predict any structural clashes. (d) Representative spectral images acquired from an unroofed membrane fragment expressing SUR1-Y1353*/Kir6.2 channels in the absence or presence of increasing concentrations of TNP-ADP. (e) Averaged spectra from (d). (f) Concentration-response relationships for binding of MgTNP-ATP (magenta) and MgTNP-ADP (cyan) to SUR1-Y1353*/Kir6.2. Combined data from multiple experiments in (f) and (g) were fit to *Equation 1* (solid lines, shaded areas represent 95% confidence). MgTNP-ATP: $E_{max}$ = 1.00 ± 0.05, $EC_{50}$ = 29.3 μM±6.5 μM, $h$ = 0.8 ± 0.1. MgTNP-ADP: $E_{max}$ = 0.99 ± 0.02, $EC_{50}$ = 4.8 μM±0.5 μM, $h$ = 0.8 ± 0.1. (g) Concentration-response relationships for binding of TNP-ATP and TNP-ADP to SUR1-Y1353*/Kir6.2 channels in the absence of $Mg^{2+}$. TNP-ATP: $E_{max}$ = 0.84 ± 0.04, $EC_{50}$ = 8.5 μM±2.4 μM, $h$ = 0.8 ± 0.1. TNP-ADP: $E_{max}$ = 0.95 ± 0.02, $EC_{50}$ = 6.1 μM±0.7 μM, $h$ = 0.8 ± 0.1. Increasing the concentration of TNP-ADP to 10 mM did not significantly change the amount of quenching (n = 3, data not shown).

DOI: https://doi.org/10.7554/eLife.41103.002

The following figure supplements are available for figure 1:

**Figure supplement 1.** ANAP fluorescence is specific to SUR-Y1353* in unroofed membrane fragments.
DOI: https://doi.org/10.7554/eLife.41103.003

**Figure supplement 2.** TNP-nucleotides are suitable for measuring nucleotide binding to $K_{ATP}$.
DOI: https://doi.org/10.7554/eLife.41103.004

**Figure supplement 3.** Mutation of the inhibitory nucleotide site in Kir6.2 does not affect binding measured at NBS2 of SUR1.
DOI: https://doi.org/10.7554/eLife.41103.005

**Figure supplement 4.** $Mg^{2+}$ effects on steady-state nucleotide binding to SUR1-Y1353*/Kir6.2.
DOI: https://doi.org/10.7554/eLife.41103.006

**Figure supplement 5.** Photobleaching correction.
DOI: https://doi.org/10.7554/eLife.41103.007

mediate $K_{ATP}$ activation by Mg-nucleotides (*Tucker et al., 1997*; *Nichols et al., 1996*). SUR1 also confers inhibition by antidiabetic sulfonylureas (SUs) and activation by $K_{ATP}$-specific $K^+$ channel openers (KCOs) (*Tucker et al., 1997*; *Nichols et al., 1996*; *Gribble et al., 1997a*; *Gribble et al., 1997b*).

Gating of $K_{ATP}$ is a complex function of the intrinsic opening and closing of the channel pore and the converging influences of the excitatory and inhibitory NBSs. Mutations that directly disrupt the NBSs or their ability to transduce nucleotide occupancy to the pore result in diseases of insulin secretion (*Quan et al., 2011*; *Ashcroft et al., 2017*). It is crucial to our understanding of $K_{ATP}$ to be able to independently measure (i) the occupancy of each NBS in real time, (ii) the effect each NBS has on $K_{ATP}$ conformation, and (iii) how binding to each NBS affects channel open probability ($P_{open}$). Electrophysiology, radioligand binding, and ATPase assays have provided rich mechanistic insight into $K_{ATP}$ gating. However, the ultimate readout of such studies is a function of nucleotide binding to all three NBSs as well as the conformational equilibria affected by binding. Photoaffinity labeling studies enable binding to each NBS to be separated, and some determination of their affinity and specificity, but require partially purified proteins, are performed over long incubation times, and involve irreversible covalent labeling (*Matsuo et al., 1999*).

An ideal method to separate the nucleotide interactions at each class of site on $K_{ATP}$ would have high spatial (specific for a single NBS) and temporal resolution, would require only small amounts of protein, could be used in a native environment (i.e. the cell or plasma membrane), and would operate under conditions compatible with electrophysiological experiments. We have developed a novel approach for studying ligand binding that fits these criteria. $K_{ATP}$ was labeled with the fluorescent non-canonical amino acid L-3-(6-acetylnaphthalen-2-ylamino)-2-aminopropanoic acid (ANAP, *Figure 1b*) and binding of fluorescent, trinitrophenyl (TNP, *Figure 1b*) nucleotide derivatives was measured using FRET (*Chatterjee et al., 2013*). This enabled us to parse the individual contribution of NBS2 to nucleotide binding/channel gating and has provided insight into the mechanism by which Mg-nucleotides affect the conformation of SUR1, and the effect this conformational change has on $P_{open}$.

## Results

### Nucleotide binding to NBS2

Binding of nucleotides to NBS2 is believed to stimulate $K_{ATP}$ activation (*Nichols et al., 1996*; *Ueda et al., 1999*). ANAP was introduced at position Y1353 (Y1353*, *Figure 1c*) of NBS2 using a transfectable amber codon suppression system (*Chatterjee et al., 2013*). Briefly, a plasmid encoding SUR1 with an amber (TAG) stop codon corresponding to position 1353 (SUR1-Y1353$^{stop}$) was co-transfected with an additional plasmid (pANAP) encoding an ANAP-specific tRNA/tRNA synthetase pair. When cells were cultured in the presence of ANAP, full-length, ANAP-labeled SUR1 protein (SUR1-Y1353*) was produced (*Figure 1—figure supplement 1d*). Fluorescence emission spectra of tagged channels were acquired from 'unroofed' membrane fragments expressing SUR1-Y1353*/Kir6.2 (*Figure 1d,e*; *Figure 1—figure supplement 1a*) (*Heuser, 2000*; *Usukura et al., 2012*; *Zagotta et al., 2016*).

In the nucleotide-bound structures of $K_{ATP}$, the aromatic ring of Y1353 forms a π-stacking interaction with the adenine ring of ADP (*Figure 1c*) (*Lee et al., 2017*; *Wu et al., 2018*). To probe nucleotide binding to $K_{ATP}$, we measured FRET between SUR1-Y1353*/Kir6.2 and fluorescent TNP-nucleotides (*Figure 1d,e*). The absorbance spectra of TNP nucleotides overlap with the emission spectrum of SUR1-Y1353* (*Figure 1—figure supplement 2a*), making them suitable FRET partners. *Figure 1—figure supplement 2b* shows the calculated distance dependence of FRET between SUR1-Y1353* and TNP-nucleotides. This steep distance dependence provides the spatial resolution necessary to discriminate between nucleotides bound directly at NBS2 (FRET efficiency close to 100%) or at other sites.

TNP-nucleotides have been used previously to probe nucleotide binding to several ABC proteins including NBD1 of SUR2A, as well as the cytoplasmic domain of Kir6.2 (*López-Alonso et al., 2012*; *Oswald et al., 2008*; *Vanoye et al., 2002*). MgTNP-ADP was able to activate $K_{ATP}$ formed by wild-type SUR1 with a mutated Kir6.2 (Kir6.2-G334D) in inside-out patches (*Figure 1—figure supplement 2c,d*; *Table 1*). G334D disrupts inhibitory nucleotide binding to Kir6.2, allowing activation to be measured in isolation (*Drain et al., 1998*; *Proks et al., 2010*).

FRET between SUR1-Y1353*/Kir6.2 and bound TNP-ADP produced a concentration-dependent decrease (quenching) of the donor ANAP peak at 470 nm and a concomitant increase in the TNP-ADP fluorescence at ~565 nm (*Figure 1d,e*). We quantified nucleotide binding using the decrease in ANAP fluorescence, as this signal was specific to $K_{ATP}$ (*Figure 1—figure supplement 1b,c*) and we observed non-specific association of TNP-nucleotides in sham-transfected membranes at high concentrations (*Figure 1—figure supplement 2e*). No change in fluorescence was observed when saturating concentrations of MgATP or MgADP were applied to SUR1-Y1353*/Kir6.2 channels (*Figure 1—figure supplement 2f*), indicating that quenching of the ANAP signal results from FRET and not an allosteric change in the environment surrounding Y1353* caused by nucleotide binding.

*Figure 1f* shows concentration-response relationships for binding of MgTNP-ATP and MgTNP-ADP to SUR1-Y1353*/Kir6.2 channels. Both nucleotides quenched ANAP completely, consistent with direct binding of TNP-nucleotides at NBS2. MgTNP-ADP bound with ~7 fold lower $EC_{50}$ than MgTNP-ATP (*Table 2*), consistent with the lower $EC_{50}$ for activation of SUR1/Kir6.2-G334D channels by MgADP vs. MgATP (7.7 μM vs. 112 μM) (*Proks et al., 2010*).

Quite unexpectedly, we observed binding of both TNP-ATP and TNP-ADP to NBS2 in the *absence* of $Mg^{2+}$, that is in 1 mM EDTA (*Figure 1g*). Both nucleotides bound with similar affinities in the absence of $Mg^{2+}$, but neither fully quenched SUR1-Y1353*/Kir6.2 fluorescence (*Table 2*). Although quenching was not 100%, the amount of FRET obtained in the absence of $Mg^{2+}$ is still consistent with nucleotides binding at NBS2. The difference in FRET efficiency at saturating nucleotide

**Table 1.** Mean ± SEM from fits of equation 5 to individual electrophysiological experiments.

| Construct | Nucleotide | $EC_{50}$ (μM) | h | n |
|---|---|---|---|---|
| SUR1/Kir6.2-G334D | MgTNP-ADP | 20.6 ± 0.4 | 1.1 ± 0.2 | 12 |
| SUR1-T1397*/Kir6.2-G334D | MgTNP-ADP | 7.3 ± 1.2 | 1.0 ± 0.1 | 5 |
| | MgADP | 44.4 ± 14.5 | 1.5 ± 0.3 | 5 |

DOI: https://doi.org/10.7554/eLife.41103.008

**Table 2.** Mean ± SEM from fits of equation 1 to individual nucleotide binding experiments.

| Construct | Nucleotide | $E_{max}$ | $EC_{50}$ (µM) | $h$ | $n$ |
|---|---|---|---|---|---|
| SUR1-Y1353*/Kir6.2 | MgTNP-ADP | 0.99 ± 0.01 | 5.5 ± 0.9 | 0.77 ± 0.03 | 8 |
| | MgTNP-ATP | 0.99 ± 0.02 | 37.5 ± 11.9 | 0.86 ± 0.07 | 7 |
| | TNP-ADP | 0.95 ± 0.01 | 6.8 ± 1.3 | 0.79 ± 0.04 | 7 |
| | TNP-ATP | 0.83 ± 0.03 | 12.8 ± 5.6 | 0.88 ± 0.06 | 7 |
| SUR1-Y1353*,K2A/Kir6.2 | MgTNP-ADP | 0.97 ± 0.03 | 18.6 ± 8.6 | 0.97 ± 0.07 | 6 |
| | MgTNP-ATP | 0.99 ± 0.02 | 31.4 ± 9.1 | 1.1 ± 0.3 | 6 |
| | TNP-ADP | 0.92 ± 0.02 | 5.0 ± 1.0 | 0.88 ± 0.05 | 6 |
| | TNP-ATP | 0.90 ± 0.03 | 8.1 ± 3.8 | 0.98 ± 0.13 | 6 |
| SUR1-T1397*/Kir6.2 | MgTNP-ADP | 1.01 ± 0.005 | 10.8 ± 0.7 | 0.98 ± 0.05 | 6 |
| | MgTNP-ATP | 1.00 ± 0.01 | 17.1 ± 3.9 | 1.0 ± 0.1 | 10 |
| | TNP-ADP | 0.98 ± 0.02 | 6.3 ± 2.3 | 0.84 ± 0.13 | 9 |
| | TNP-ATP | 0.94 ± 0.01 | 4.7 ± 0.8 | 0.83 ± 0.09 | 8 |
| SUR1-Y1353*/Kir6.2-G334D | MgTNP-ADP | 1.01 ± 0.04 | 8.2 ± 1.9 | 0.90 ± 0.15 | 5 |

DOI: https://doi.org/10.7554/eLife.41103.009

concentrations in the presence and absence of $Mg^{2+}$ may reflect a change in orientation between the donor and acceptor, as the NBDs are expected to change conformation in the presence of $Mg^{2+}$ (*Lee et al., 2017*; *Martin et al., 2017a*).

Despite the relatively close proximity of NBS1 to Y1353* (39 Å to TNP-ATP aligned at NBS1, *Figure 1c*), the FRET we observed was primarily the result of quenching by TNP-ATP/ADP bound at NBS2. When nucleotides are bound directly to NBS2, FRET efficiency is very high and any nucleotide binding at NBS1 would be expected to have little or no effect on our measurements (*Corry et al., 2005*). Nucleotides at NBS1 would only contribute to the observed FRET signal if NBS2 were unoccupied. However, with no nucleotide bound to NBS2, the NBDs would not dimerize and Y1353* would be ~56 Å away from TNP-nucleotides bound to NBS1 (based on PDB 6BAA) (*Martin et al., 2017a*). At this distance, the predicted FRET efficiency is <20% (*Figure 1—figure supplement 2b*), much less that the FRET efficiencies we observed for TNP nucleotide quenching ±$Mg^{2+}$ (*Figure 1f,g*; *Table 2*).

In the Mg-nucleotide-bound 'quatrefoil' structure of $K_{ATP}$, the inhibitory NBS of Kir6.2 is within ~33 Å of Y1353 (*Figure 1—figure supplement 3a*) (*Lee et al., 2017*). To eliminate a possible contribution of nucleotide bound at Kir6.2 to our measured FRET signal, we measured binding of MgTNP-ADP to SUR1-Y1353* co-expressed with Kir6.2-G334D (*Figure 1—figure supplement 3b*). The $EC_{50}$ was very similar to that measured with SUR1-Y1353*/Kir6.2 (*Table 2*), suggesting that either the NBS on Kir6.2 was too distant for FRET with Y1353*, or that appreciable binding to Kir6.2 did not occur at concentrations at which NBS2 was unoccupied.

## $Mg^{2+}$ locks nucleotides at NBS2

If nucleotides can bind NBS2 in the absence of $Mg^{2+}$, why is $K_{ATP}$ only activated in the presence of $Mg^{2+}$? The simplest explanation is that binding to NBS2 is $Mg^{2+}$-independent, but $Mg^{2+}$ is required to support the conformational change that promotes channel activation. Presumably, this conformational change involves NBD dimerization, as observed in the cryo-EM structures of $K_{ATP}$ (*Lee et al., 2017*; *Wu et al., 2018*). Because nucleotide dissociation would require opening of the NBD dimer, dimerization is expected to slow the nucleotide off rate. We therefore measured the time course of nucleotide dissociation from SUR1-Y1353*/Kir6.2 in the presence and absence of $Mg^{2+}$.

In the absence of $Mg^{2+}$, TNP-ADP dissociated very rapidly from SUR1-Y1353*/Kir6.2 (*Figure 2a, e*; *Table 3*). Dissociation of TNP-ADP was greatly slowed by the presence of $Mg^{2+}$ (p=0.001). Similar, but less dramatic results were obtained with TNP-ATP (*Figure 2b,e*; p=0.0001; *Table 3*). Whereas the time courses of dissociation of TNP-ADP ±$Mg^{2+}$ and TNP-ATP in the presence of $Mg^{2+}$ were well described by single exponential decays, the dissociation of TNP-ATP in the absence of $Mg^{2+}$ appeared bi-exponential.

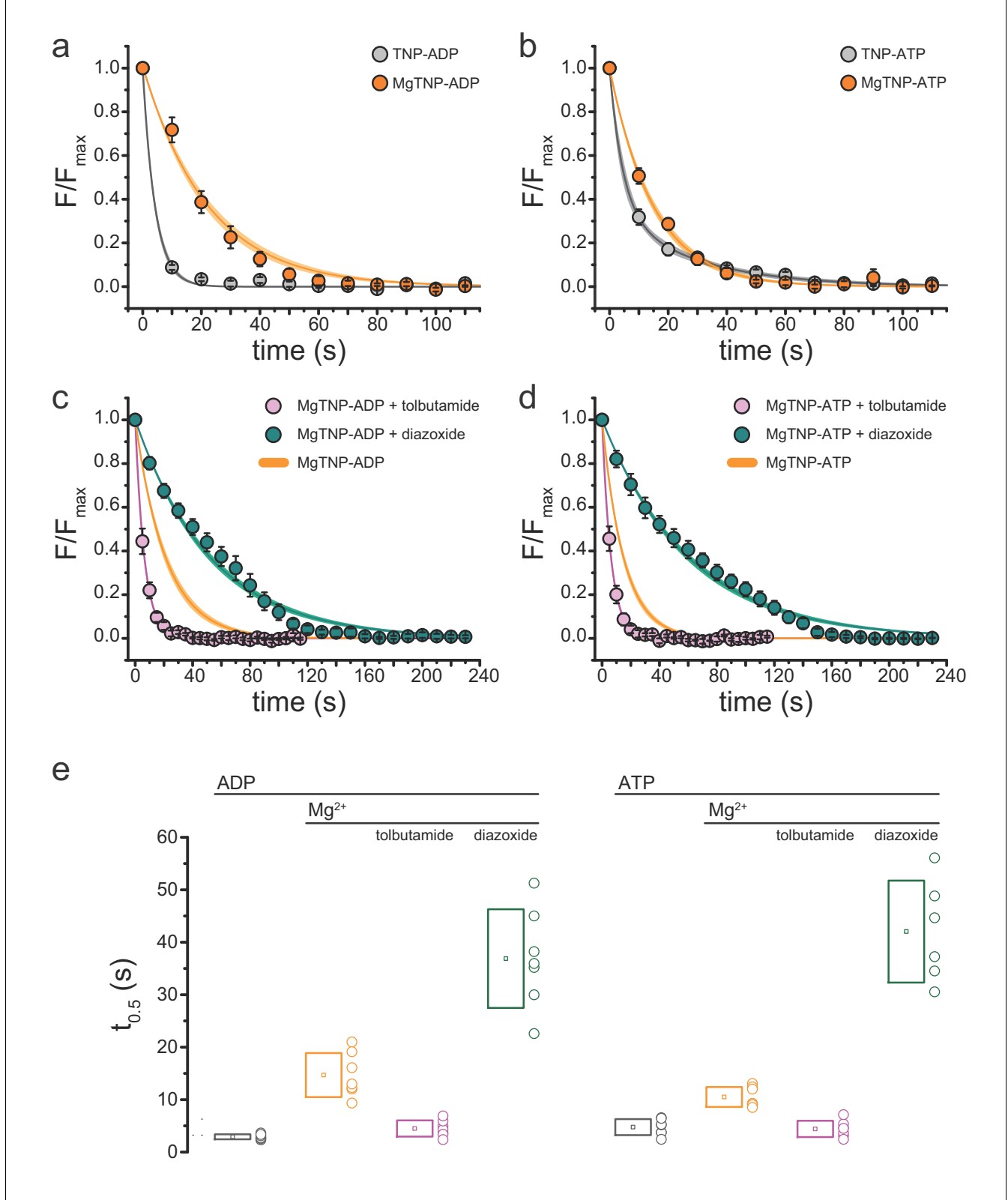

**Figure 2.** Conditions that favor channel opening slow nucleotide dissociation from NBS2. (**a**) Average dissociation time course of TNP-ADP from SUR1-Y1353*/Kir6.2 $K_{ATP}$ channels in the presence (orange) and absence (gray) of $Mg^{2+}$. TNP-ADP: $\tau$ = 4.2 s±0.2 s; MgTNP-ADP: $\tau$ = 21.3 s±0.8 s. (**b**) Dissociation time course of TNP-ATP from SUR1-Y1353*/Kir6.2 channels in the presence and absence of $Mg^{2+}$. TNP-ATP: $\tau_1$ = 4.5 s±0.9 s, $A_1$ = 0.65 ± 0.06, $\tau_2$ = 28.2 s±3.7 s, $A_2$ = 0.35 ± 0.06; MgTNP-ATP: $\tau$ = 15.0 s±0.5 s. (**c,d**) Dissociation time course of MgTNP-ADP (**c**) or MgTNP-ATP (**d**)

*Figure 2 continued on next page*

*Figure 2 continued*

from SUR1-Y1353*/Kir6.2 in the presence of 500 µM tolbutamide (pink) or 340 µM diazoxide (teal). (c) The orange curve is the fit to the MgTNP-ADP data from (a). Tolbutamide: $\tau$ = 6.4 s±0.2 s; diazoxide: $\tau$ = 52.6 s±1.4 s. (d) The orange curve is the fit to MgTNP-ATP data from (b). Tolbutamide: $\tau$ = 6.3 s±0.1 s; diazoxide: $\tau$ = 60.5 s±1.4 s. (e) Time to 50% dissociation ($t_{0.5}$) for the individual fits to the data shown in (a)-(d) for better comparison between single-exponential and more complex time courses. Boxes represent mean ±SD. Individual data points are shown to the right of the boxes. Combined data from multiple experiments in (a)-(d) were fit to single exponential decays (*Equation 2*). Dissociation of TNP-ATP in the absence of $Mg^{2+}$ was better fit with a bi-exponential decay (*Equation 3*).

DOI: https://doi.org/10.7554/eLife.41103.010

Our data are consistent with an activation model for $K_{ATP}$ in which the NBDs of SUR1 dimerize in the presence of $Mg^{2+}$, preventing rapid nucleotide dissociation. However, it is formally possible that $Mg^{2+}$ could directly stabilize nucleotide binding without any conformational change in the NBDs. To test this hypothesis, we examined nucleotide dissociation rates in the presence of the sulfonylurea (SU) tolbutamide (*Figure 2c,d,e*, *Table 3*). The cryo-EM structure of Kir6.2/SUR1 in the presence of glibenclamide suggests that SUs prevent NBD dimerization, and electrophysiological studies further demonstrate that SUs prevent Mg-nucleotide induced current activation (*Martin et al., 2017a*; *Proks et al., 2014*). In the presence of tolbutamide, the dissociation rates for MgTNP-ADP and MgTNP-ATP were not significantly different from those measured for nucleotides in the absence of both tolbutamide and $Mg^{2+}$ (*Figure 2e*, *Table 3*; p=0.1, for TNP-ADP, p=1 for TNP-ATP). This argues that the slower off rate for TNP-ADP and TNP-ATP in the presence of $Mg^{2+}$ must result from NBD dimerization, not a direct stabilization of nucleotide binding by $Mg^{2+}$.

As further support, we tested the effect of diazoxide (a KCO) on the nucleotide dissociation rate. It has been proposed that nucleotide binding at the NBDs of SUR2A stabilizes binding of the KCO pinacidil (*Gribble et al., 2000*). The time course of nucleotide dissociation in the presence of diazoxide did not closely follow a single exponential decay, but the drug greatly reduced the dissociation rates for both MgTNP-ADP and MgTNP-ATP (*Figure 2c,d,e*; *Table 3*). This suggests that KCOs function by stabilizing the activated (NBD-dimerized) conformation of SUR1.

In the absence of any conformational change (i.e. in $Mg^{2+}$-free solution), nucleotide binding to NBS2 would be expected to follow a simple binding equilibrium (Scheme 1), with $K_d = k_{off}/k_{on}$.

$$\text{TNP} - \text{nucleotide} + \text{SUR1} \underset{k_{off}}{\overset{k_{on}}{\rightleftharpoons}} \text{TNP} - \text{nucleotide} \cdot \text{SUR1} \qquad \text{(Scheme 1)}$$

A simple activation model like the one proposed by Del Castillo and Katz for acetylcholine receptors (Scheme 2), in which nucleotides bind SUR1 and change its conformation can be invoked to explain the decrease in the apparent rate of nucleotide dissociation (*Del Castillo and Katz, 1957*). The nucleotide dissociation rate would depend on the kinetics of entry into ($\alpha$) and exit from ($\beta$) the

**Table 3.** Mean ± SD from exponential fits to individual experiments measuring nucleotide wash-out time courses.

| SUR1 construct | Nucleotide | Drug | $t_{0.5}$ (s) | n |
|---|---|---|---|---|
| Y1353* | TNP-ADP | | 2.9 ± 0.5 | 7 |
| | MgTNP-ADP | | 14.7 ± 4.2 | 7 |
| | MgTNP-ADP | tolbutamide | 4.5 ± 1.5 | 7 |
| | MgTNP-ADP | diazoxide | 36.9 ± 9.4 | 7 |
| | TNP-ATP | | 4.8 ± 1.5 | 7 |
| | MgTNP-ATP | | 10.5 ± 1.9 | 7 |
| | MgTNP-ATP | tolbutamide | 4.4 ± 1.6 | 7 |
| | MgTNP-ATP | diazoxide | 42.0 ± 9.7 | 6 |
| Y1353*,K2A | TNP-ADP | | 3.8 ± 1.0 | 3 |
| | MgTNP-ADP | | 9.3 ± 2.1 | 4 |

DOI: https://doi.org/10.7554/eLife.41103.011

activated state (SUR1', in which the NBDs are dimerized), as well as the intrinsic $k_{off}$. In Scheme 2, $EC_{50,apparent} = K_d/(\alpha/\beta +1)$.

$$\text{MgTNP} + \text{SUR1} \underset{k_{off}}{\overset{k_{on}}{\rightleftharpoons}} \text{MgTNP} \cdot \text{SUR1} \underset{\beta}{\overset{\alpha}{\rightleftharpoons}} \text{MgTNP} \cdot \text{SUR1}' \qquad \text{(Scheme 2)}$$

In our experiments, the off-rate for TNP-ADP was reduced ~5 fold in the presence of $Mg^{2+}$, whereas the $EC_{50}$ for TNP-ADP was largely unaffected (*Figure 2e*, *Figure 1—figure supplement 4a*, *Table 2*, *Table 3*). $Mg^{2+}$ reduced the off-rate for TNP-ATP by a factor of 2.2, but actually increased the $EC_{50}$ for binding by a factor of ~3 (*Figure 2e*, *Figure 1—figure supplement 4b*, *Table 2*, *Table 3*). In the limiting case for Scheme 2, in which β>>α, the apparent $EC_{50}$ would be equal to the actual $K_d$. (i.e. that measured in $Mg^{2+}$-free solution). Therefore, the concentration-response curve for nucleotide binding would not be expected to shift to the right in the presence of $Mg^{2+}$ through a change in activation alone. As we observed such a shift for Mg-TNP-ATP, we conclude that in addition to stabilizing the NBDs in a dimerized state when nucleotides are present, $Mg^{2+}$ must also decrease the intrinsic on-rate for nucleotide binding ($k_{on}$).

The nucleotide dissociation rates measured in the presence of $Mg^{2+}$ are slower than the deactivation rates observed for SUR1/Kir6.2-G334D currents following nucleotide washout (*Proks et al., 2010*). However, it should be noted that current decay reflects a conformational change in the channel and not the rate of nucleotide dissociation, and may be affected by the presence of four SUR1 monomers per channel. Furthermore, the deactivation rates were measured for MgATP/ADP and not TNP-nucleotides. Ultimately, a direct comparison of TNP-nucleotide dissociation rates and deactivation rates will require simultaneous measurement of current and binding.

## Mutation of the Walker$_A$ lysine of NBS2 affects the conformational change in SUR1

We next sought to explore the mechanistic consequences of mutating the Walker$_A$ nucleotide binding motif of NBS2. In other ABC proteins, and in the Mg-nucleotide-bound structure of K$_{ATP}$, the Walker$_A$ lysine coordinates the β- and γ-phosphates of bound nucleotides (*Figure 3a*). Previous electrophysiological studies showed that mutation of the Walker$_A$ lysine in NBS2 to alanine (K1384A, K2A) reduced the maximal activation of SUR1/Kir6.2-G334D currents by MgATP and shifted the concentration dependence of MgADP activation >20 fold to higher concentrations, such that the maximal response could not be determined (*Proks et al., 2014*). This is consistent with the K2A mutation affecting channel activation.

We mutated the Walker$_A$ lysine of NBS2 to alanine in channels labeled with ANAP at NBS2 (SUR1-Y1353*,K2A/Kir6.2) and measured nucleotide binding with and without $Mg^{2+}$ (*Figure 3*). In the absence of $Mg^{2+}$, we observed no significant difference in steady-state TNP-ATP and TNP-ADP binding to SUR1-Y1353*,K2A/Kir6.2 compared to SUR1-Y1353*/Kir6.2 (*Figure 3b,c*; *Table 2*), suggesting the K2A mutation has no effect on the intrinsic nucleotide affinity in the absence of $Mg^{2+}$. The apparent affinity for MgTNP-ATP was also unaffected by the K2A mutation (*Figure 3d*; *Table 2*). However the K2A mutation disrupted MgTNP-ADP binding ($EC_{50}$ = 18.6 μM for K2A vs. 4.8 μM for SUR1-Y1353*, p=0.04; *Figure 3e*, *Table 2*). Similar to SUR1-Y1353*/Kir6.2 channels, $Mg^{2+}$ caused a rightward shift in the concentration dependence of nucleotide binding to SUR1-Y1353*,K2A/Kir6.2 (*Figure 3—figure supplement 1*).

As we observed no change in the TNP-ADP concentration-response in the absence of $Mg^{2+}$, the K2A mutation may affect MgTNP-ADP binding indirectly, by reducing the ability of the nucleotide to cause NBD dimerization. To explore this idea further, we measured the rate of MgTNP-ADP dissociation from SUR1-Y1353*,K2A/Kir6.2 (*Figure 3f*, *Table 3*). The K2A mutation increased the off-rate for MgTNP-ADP compared to SUR1-Y1353*/Kir6.2 channels (*Table 3*; p=0.04). There was no apparent change in the off-rate for TNP-ADP in the absence of $Mg^{2+}$, compared to SUR1-Y1353*/Kir6.2 (p=0.8). Taken together, these data suggest that the K2A mutation destabilized NBD dimerization induced by MgTNP-ADP. However, these results do not rule out the possibility that the K2A mutation also affected the intrinsic on-rate for MgTNP-ADP.

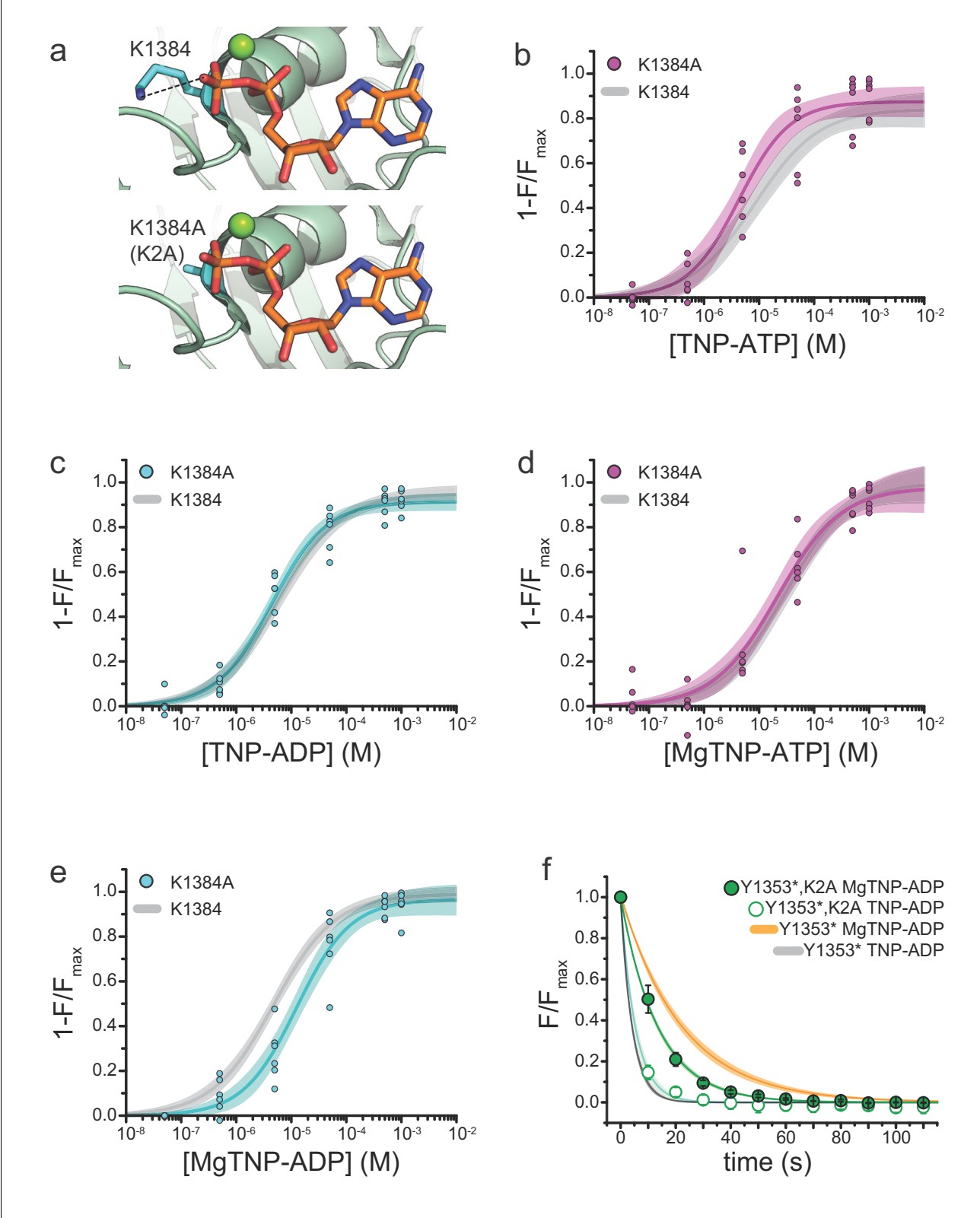

**Figure 3.** Mutation of the Walker$_A$ motif of NBS2 affects apparent the affinity for MgTNP-ADP. (a) Interaction of the Walker$_A$ lysine (K1384) with MgADP bound to NBS2 of SUR1 (from PDB accession # 6C3O) (*Lee et al., 2017*). The K1384A (K2A) mutation (modeled *below*) disrupts this interaction. (b) Binding of TNP-ATP to SUR1-Y1353*,K2A/Kir6.2 K$_{ATP}$ channels in the absence of Mg$^{2+}$. Combined data from multiple experiments in (b)-(e) were fit to *Equation 1*. $E_{max}$ = 0.87 ± 0.03, $EC_{50}$ = 4.3 μM±0.1 μM, *h* = 0.9 ± 0.2. Gray curves are fits to SUR1-Y1353*/Kir6.2 channels from *Figure 1*. (c) Binding of

*Figure 3 continued on next page*

*Figure 3 continued*

TNP-ADP to SUR1-Y1353*,K2A/Kir6.2 in the absence of $Mg^{2+}$. $E_{max}$ = 0.91 ± 0.02, $EC_{50}$ = 4.4 µM±0.1 µM, $h$ = 0.9 ± 0.1. (d) Binding of MgTNP-ATP to SUR1-Y1353*,K2A/Kir6.2. $E_{max}$ = 0.97 ± 0.06, $EC_{50}$ = 21.5 µM±6.7 µM, $h$ = 0.8 ± 0.1. (e) Binding of MgTNP-ADP to SUR1-Y1353*,K2A/Kir6.2. $E_{max}$ = 0.96 ± 0.04, $EC_{50}$ = 12.5 µM±2.5 µM, $h$ = 0.9 ± 0.1. (f) Dissociation time course of MgTNP-ADP (green, filled symbols) and TNP-ADP (green, open symbols) from SUR1-Y1353*,K2A/Kir6.2. Combined data from multiple experiments were fit to a single exponential; τ = 13.5 s±0.3 s (for MgTNP-ADP) and τ = 5.4 s±0.8 s (for TNP-ADP). Fits to the dissociation of TNP-ADP from SUR1-Y1353*/Kir6.2 in the presence (orange) and absence (gray) of $Mg^{2+}$ from *Figure 2* are shown for comparison.

DOI: https://doi.org/10.7554/eLife.41103.012

The following figure supplement is available for figure 3:

**Figure supplement 1.** $Mg^{2+}$ effects on steady-state nucleotide binding to SUR1-Y1353*,K2A/Kir6.2.

DOI: https://doi.org/10.7554/eLife.41103.013

## Correlation of binding with gating

We next sought to correlate nucleotide binding with changes in channel activity. Despite our ability to demonstrate a $Mg^{2+}$- and nucleotide-dependent conformational change in SUR1-Y1353*/Kir6.2, we were unable to measure activation of SUR1-Y1353*/Kir6.2 channels by Mg-nucleotides. There are two possible explanations for this observation. The first is that SUR1-Y1353* disrupts communication between the NBD-dimerized SUR1 and Kir6.2. Inspection of the Mg-nucleotide-bound 'quatrefoil' structure shows that position 1353 is adjacent to residues (e.g. R1352) at the interface between SUR1 and Kir6.2 (*Lee et al., 2017*). Mutation of R1352 disrupts $K_{ATP}$ activation (*de Wet et al., 2012*; *Magge et al., 2004*). Another possible explanation for the lack of activation of SUR1-Y1353*/Kir6.2 currents is that channels containing SUR1 subunits truncated at Y1352 (i.e. no ANAP incorporated) traffic to the plasma membrane. These channels would not be activated by nucleotides as they lack a complete NBS2. $K_{ATP}$ channels with SUR1 truncated at residue 1330 at the start of NBD2, traffic to the plasma membrane but are not activated by MgADP (*Sakura et al., 1999*).

In fluorescence measurements, we only observe full-length SUR1-Y1353*/Kir6.2 channels as only these are labeled with ANAP (*Figure 1—figure supplement 1*). However, in electrophysiological experiments it may be possible to measure currents from channels with both full-length and truncated SUR1. Using a surface expression assay, we observed no difference in the ability of full-length SUR1-Y1353* and truncated SUR1-Y1353$^{stop}$ (no ANAP included in the culture medium) to chaperone HA-tagged Kir6.2 to the plasma membrane (*Figure 4—figure supplement 1a*) (*Zerangue et al., 1999*). Truncated SUR1-Y1353$^{stop}$/Kir6.2 (expressed without pANAP) formed functional channels that were not activated by MgADP (*Figure 4—figure supplement 1b*). 100 µM MgADP increased full-length 3xFLAG_SUR1/Kir6.2 currents 3.6-fold over the current in nucleotide-free control solutions (*Figure 4—figure supplement 1b*, *Table 4*). In contrast, 100 µM MgADP inhibited current from truncated SUR1-Y1353$^{stop}$ by 59% (*Figure 4—figure supplement 1b*, *Table 4*). This degree of inhibition is similar to the ~60% reduction in wild-type SUR1/Kir6.2 current by 100 µM ADP in $Mg^{2+}$-free solutions, in which only inhibition is expected (*Proks et al., 2010*), confirming that SUR1-Y1353$^{stop}$/Kir6.2 channels failed to activate. Currents from SUR1-Y1353$^{stop}$/Kir6.2 co-expressed with pANAP alone (66% full-length SUR1-Y1353* expected, *Figure 1—figure supplement 1d*) or with pANAP and a dominant-negative ribosomal release factor peRF1-E55D (92% full-length SUR-Y1353* expected, *Figure 1—figure supplement 1d*) were inhibited by 100 µM MgADP to a similar extent as currents from cells expressing only truncated channels (*Figure 4—figure supplement 1b*, *Table 4*) (*Schmied et al., 2014*). This suggests that SUR-Y1353* fails to support nucleotide activation. To confirm this, we also looked at currents in patches excised from cells expressing

**Table 4.** Mean ± SD response to 100 µM MgADP relative to the current in nucleotide-free solution.

| Construct | ANAP | pANAP | peRF1-E55D | Nucleotide | $I/I_{control}$ | n |
|---|---|---|---|---|---|---|
| 3xFLAG_SUR1/Kir6.2 | – | – | – | 100 µM MgATP | 3.6 ± 1.3 | 13 |
| SUR1-Y1353$^{stop}$/Kir6.2 | + | – | – | 100 µM MgATP | 0.41 ± 0.7 | 7 |
| SUR1-Y1353$^{stop}$/Kir6.2 | + | + | – | 100 µM MgATP | 0.42 ± 0.05 | 5 |
| SUR1-Y1353$^{stop}$/Kir6.2 | + | + | + | 100 µM MgATP | 0.37 ± 0.10 | 3 |

DOI: https://doi.org/10.7554/eLife.41103.014

SUR1-Y1353*/Kir6.2-G334D (co-expressed with pANAP and pERF1-E55D). Wild-type SUR1 paired with Kir6.2-G334D produced channels that were activated by MgADP and inhibited by tolbutamide (*Figure 4—figure supplement 1c*). When measured under the same conditions, SUR1-Y1353*/Kir6.2 was inhibited by tolbutamide, but we observed no activation by MgADP.

We therefore sought to label an additional position near NBS2 of SUR1 that produces functional, nucleotide-activated channels when paired with Kir6.2. We mutated the codons for 34 different residues (amino acids 1386–1397, 1400–1408, and 1410–1423) to the amber stop codon and screened the constructs for their ability to promote surface expression of HA-Kir6.2 in the presence of ANAP. We focused on SUR1-T1397[stop], as this construct strongly increased the surface expression of HA-Kir6.2 in the presence of ANAP (*Figure 4—figure supplement 1a*; p=0.005). Further, in the absence of ANAP, this construct decreased the surface expression of HA-Kir6.2 (p=0.009) compared to no-SUR1 controls (*Figure 4—figure supplement 1a*), suggesting that channels with SUR truncated at this position may be selectively retained within the cell. Western blots of total protein from cells expressing SUR1-T1397[stop] in the presence of ANAP and pANAP/peRF1-E55D confirmed that >95% of SUR1-T1397* was full length (*Figure 4—figure supplement 1d*). The expected distance between T1397* and TNP-nucleotides bound to NBS2 is ~18.3 Å (*Figure 4a*), close enough for >99% FRET efficiency (*Figure 1—figure supplement 2b*).

Binding of MgTNP-ADP and MgTNP-ATP to SUR1-T1397*/Kir6.2 (*Figure 4b*) was qualitatively similar to SUR1-Y1353*/Kir6.2 (*Figure 1*, *Table 2*), although there was no obvious difference in binding affinity between MgTNP-ADP and MgTNP-ATP for SUR1-T1397*/Kir6.2. Both TNP-ADP and TNP-ATP bound to SUR1-T1397*/Kir6.2 in the absence of $Mg^{2+}$ (*Figure 4c*; *Table 2*) as we observed for SUR1-Y1353*/Kir6.2. $Mg^{2+}$ caused a rightward shift in the concentration dependence of TNP-ATP and TNP-ADP binding (*Figure 4—figure supplement 2*).

In excised patches from cells expressing SUR1-T1397*/Kir6.2, we measured robust currents that were inhibited by ATP (*Figure 4—figure supplement 1e*). Interestingly, the apparent affinity for ATP inhibition of SUR1-T1397*/Kir6.2 was lower than for wild-type $K_{ATP}$ ($IC_{50}$ was 53.9 μM vs. 11.2 μM, respectively; *Table 5*). More importantly, SUR1-T1397* supported activation of Kir6.2-G334D by both MgADP (*Figure 4d*, *Table 1*) and MgTNP-ADP (*Figure 5a,b*; *Table 1*). The $EC_{50}$ for channel activation by MgTNP-ADP (determined from Hill fits) was 7.3 μM ± 1.2 μM as compared to the $EC_{50}$ of 10.8 μM ± 0.7 μM obtained for MgTNP-ADP binding to NBS2. Although activation (by necessity) was measured with Kir6.2-G334D, and our binding measurements were conducted with wild-type Kir6.2, we do not expect this to have affected our measurements as T1397* is the same distance (~33 Å) as Y1353* from the inhibitory binding site on Kir6.2, and expression of SUR1-Y1353* with Kir6.2-G334D did not affect binding measurements (*Figure 1—figure supplement 3b*, *Table 2*).

## Discussion

Taken together, our data suggest a modified model for SUR1/Kir6.2 $K_{ATP}$ activation by nucleotide binding to SUR1 (*Figure 6*). ADP and ATP bind to NBS2 in the absence of $Mg^{2+}$. However, without $Mg^{2+}$ as a co-factor, adenine nucleotides cannot dimerize the NBDs and initiate the conformational change in SUR1 required to affect $P_{open}$. As the NBDs fail to dimerize, nucleotides dissociate rapidly. If both $Mg^{2+}$ and nucleotides are present, the NBDs dimerize, resulting in slower nucleotide dissociation. NBD dimerization initiates a conformational change in SUR1 that ultimately communicates nucleotide occupancy to the pore, affecting $P_{open}$. Inhibitory SUs prevent NBD dimerization. Thus, nucleotides dissociate rapidly even in the presence of $Mg^{2+}$ and $P_{open}$ is not affected by nucleotide occupancy at NBS2. KCOs have the opposite effect. They stabilize NBD dimerization, prolonging nucleotide occupancy at NBS2 and channel activation.

Although our data do not offer any direct structural insight into how NBD dimerization propagates to Kir6.2, our ability to measure MgTNP-ADP binding to NBS2 of SUR1 and channel activation under similar conditions furnishes some mechanistic insights into the relationship between nucleotide binding and channel opening. We simultaneously fit the combined data sets for MgTNP-ADP binding to SUR1-T1397*/Kir6.2 and activation of SUR1-T1397*/Kir6.2-G334D currents with several gating models (*Figure 5*, *Figure 5—figure supplement 1*, *Figure 5—figure supplement 2*). For each model, equations for binding and $P_{open}$ used common values for $K_A$ (the equilibrium association constant in units of $M^{-1}$, constrained such that $K_A > 0$), $L$ (the equilibrium constant for the intrinsic opening of unliganded SUR1-T1397*/Kir6.2-G334D, fixed to 0.52 based on the measured $P_{open}$ of

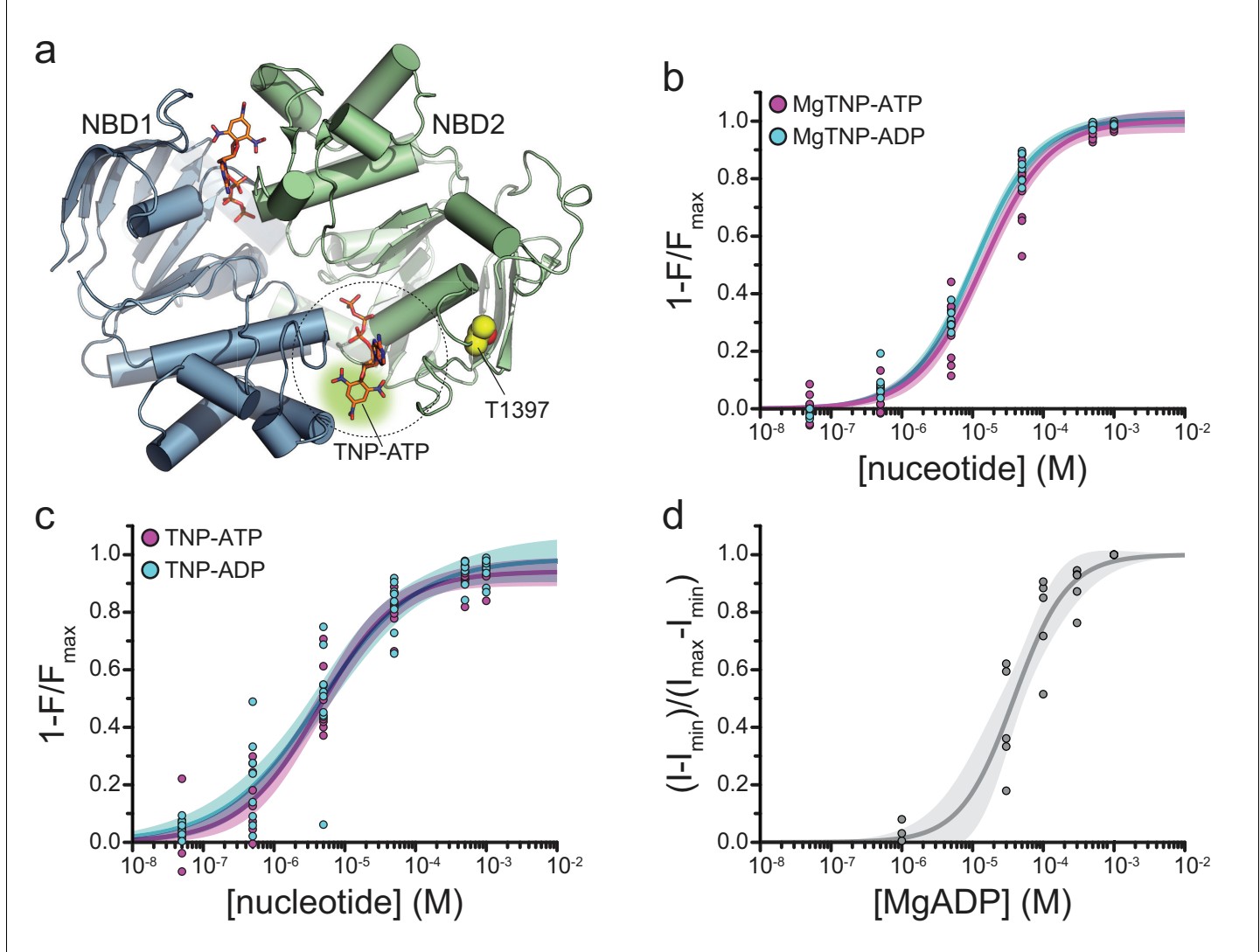

**Figure 4.** Binding of TNP-nucleotides to SUR1-T1397*/Kir6.2 K$_{ATP}$ channels. (a) Structure of the ligand-bound NBDs of SUR1 (from PDB accession # 6C3O) with TNP-ATP aligned as in *Figure 1c* (*Lee et al., 2017*). T1397 in NBD2 is highlighted in yellow to mark the placement of ANAP in our experiments. Combined data from multiple experiments in (c) and (d) were fit to *Equation 1*. (b) Concentration-response relationships for binding of MgTNP-ATP (magenta) and MgTNP-ADP (cyan) to SUR1-T1397*/Kir6.2 channels. MgTNP-ATP: $E_{max}$ = 1.00 ± 0.02, $EC_{50}$ = 14.7 µM±1.8 µM, $h$ = 1.0 ± 0.1. MgTNP-ADP: $E_{max}$ = 1.0 ± 0.01, $EC_{50}$ = 10.6 µM±0.07 µM, $h$ = 0.97 ± 0.06. (c) Concentration-response relationships for binding of TNP-ATP and TNP-ADP to SUR1-Y1397*/Kir6.2 channels in the absence of Mg$^{2+}$. TNP-ATP: $E_{max}$ = 0.94 ± 0.02, $EC_{50}$ = 4.8 µM±0.7 µM, $h$ = 0.8 ± 0.1. TNP-ADP: $E_{max}$ = 0.98 ± 0.05, $EC_{50}$ = 5.6 µM±1.7 µM, $h$ = 0.7 ± 0.1. (d) Activation of SUR1-T1397*/Kir6.2-G334D channels by MgADP in inside-out patches. Combined data from multiple experiments were fit with *Equation 5*. $EC_{50}$ = 38.6 µM±5.8 µM, $h$ = 1.2 ± 0.2.

DOI: https://doi.org/10.7554/eLife.41103.015

The following figure supplements are available for figure 4:

**Figure supplement 1.** SUR1-T1397*/Kir6.2 K$_{ATP}$ channels traffic to the plasma membrane and form functional channels.

DOI: https://doi.org/10.7554/eLife.41103.016

**Figure supplement 2.** Mg$^{2+}$ effects on steady-state nucleotide binding to SUR1-T1397*.

DOI: https://doi.org/10.7554/eLife.41103.017

SUR1/Kir6.2-G334D (*Proks et al., 2010*)), and *E* (the factor by which MgTNP-ADP binding favors activation, constrained to *E* > 1). Goodness of fit was assessed by comparing the residual sum of the squared deviations (RSS) between the data and fit curves (*Figure 5—figure supplement 2*). As K$_{ATP}$ channels open in the absence of Mg$^{2+}$ and nucleotides, we excluded any gating models that lacked unliganded opening. None of the available cryo-EM structures (*Lee et al., 2017*; *Martin et al.,*

**Table 5.** Mean ± SEM from fits of *Equation 4* to individual electrophysiological experiments.

| Construct | Nucleotide | $IC_{50}$(μM) | h | n |
|---|---|---|---|---|
| SUR1/Kir6.2 | ATP | 11.2 ± 0.1 | −1.3 ± 0.1 | 6 |
| SUR1-T1397*/Kir6.2 | ATP | 53.9 ± 9.4 | −1.2 ± 0.1 | 6 |

DOI: https://doi.org/10.7554/eLife.41103.021

*2017a*; *Wu et al., 2018*; *Li et al., 2017*; *Martin et al., 2017b*). show any direct interactions between neighboring SUR1 subunits. Therefore, we excluded models in which binding to one SUR1 directly affects binding to the other SUR1 subunits (i.e. with binding cooperativity).

With these assumptions, the best fits to the data were obtained with an extended Monod-Wyman-Changeux model (*Figure 5b*; $R^2$ = 0.98, RSS = 0.15) (*Monod et al., 1965*). In this model,

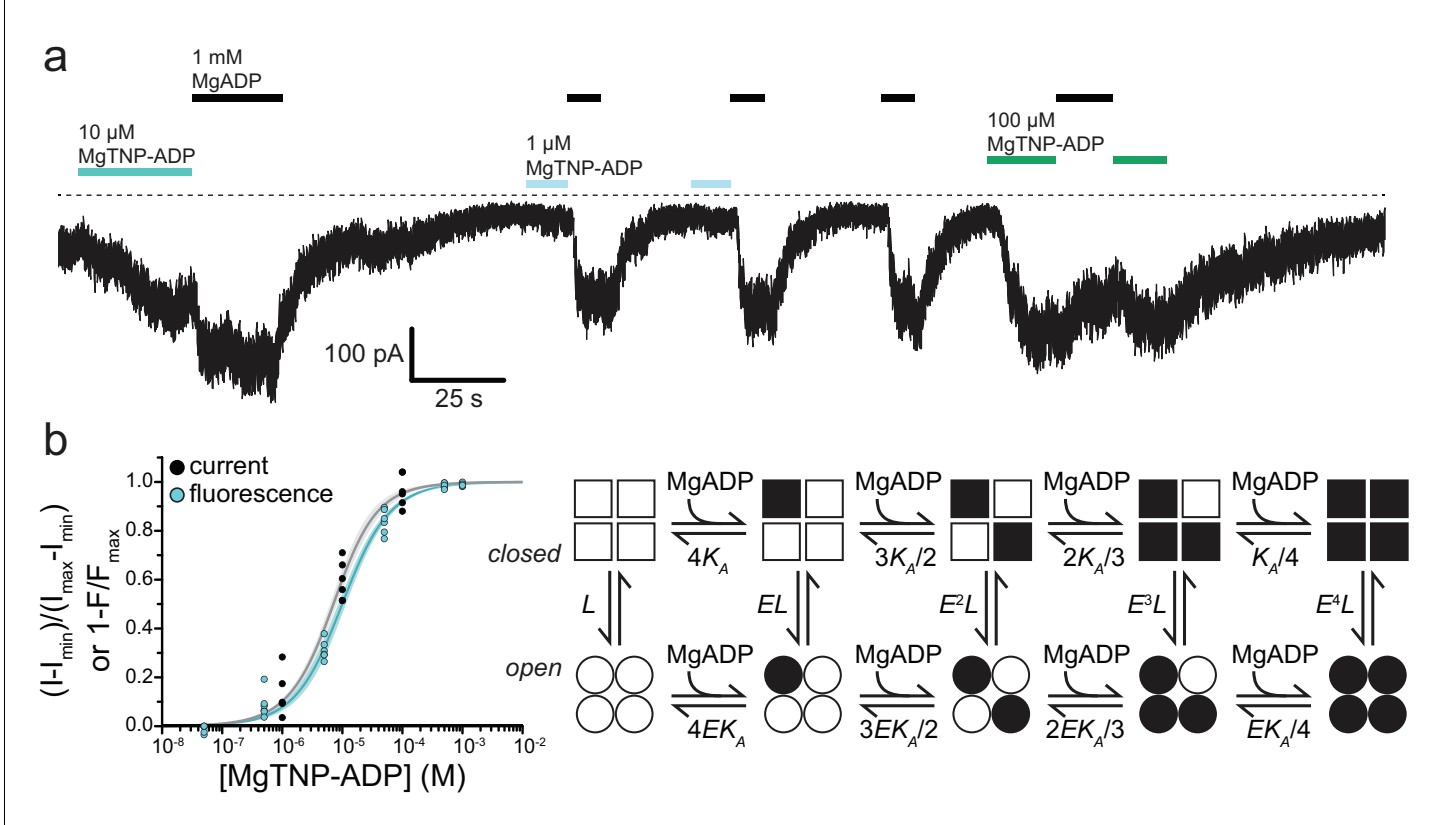

**Figure 5.** .Activation of $K_{ATP}$ by Mg-nucleotide binding at SUR1 proceeds via an MWC-type reaction. (a) Continuous current trace from an inside out patch expressing SUR1-T1397*/Kir6.2-G334D showing activation by MgADP and MgTNP-ADP. The dotted line indicates the zero-current level. Downward deflections indicate increasing current. (b) Combined data from multiple experiments for MgTNP-ADP binding (cyan from *Figure 4b*; SUR1-T1397*/Kir6.2) and current activation (gray from SUR1-T1397*/Kir6.2-G334D in separate experiments) were fit simultaneously with MWC expressions for binding (cyan, *Equation 6*) and activation (gray, *Equation 7*) with shared parameters for the two expressions. The schematic on the right describes the MWC model. Closed $K_{ATP}$ channels are drawn as squares and open channels as circles. Filled symbols represent subunits bound to MgTNP-ADP (abbreviated as MgADP). Channels open by a concerted flip of all four subunits. $L$ = 0.52, $E$ = 2.2 ± 0.2, and $K_A$ = 5.8×10$^4$ M$^{-1}$±1.0×10$^4$ M$^{-1}$. Each MgTNP-ADP binding event is independent and each MgTNP-ADP-bound NBS2 contributes equally to increase $P_{open}$.

DOI: https://doi.org/10.7554/eLife.41103.018

The following figure supplements are available for figure 5:

**Figure supplement 1.** Fits to alternate gating models.
DOI: https://doi.org/10.7554/eLife.41103.019

**Figure supplement 2.** Residuals of fits to alternate gating models.
DOI: https://doi.org/10.7554/eLife.41103.020

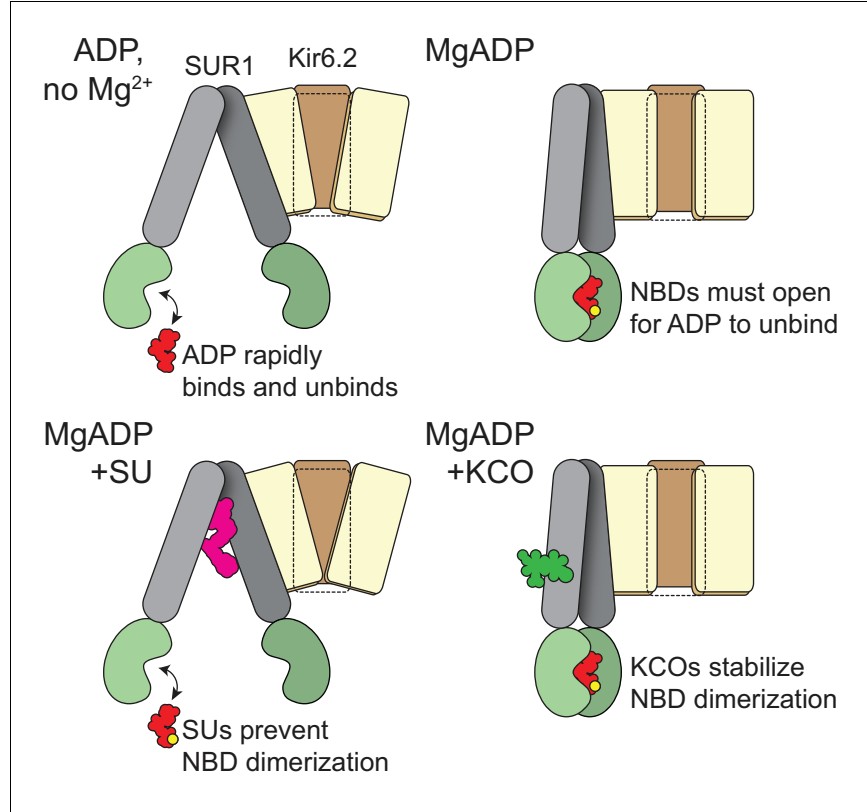

**Figure 6.** Schematic for activation of $K_{ATP}$ by nucleotide binding to *SUR1*. (*top, left*). In the absence of $Mg^{2+}$ nucleotides bind and rapidly dissociate from the NBSs of SUR1, without affecting its conformation. Nucleotide occupancy of SUR1 is not communicated to Kir6.2. (*top, right*) In the presence of both $Mg^{2+}$ and nucleotide, the NBD dimer interface is stabilized. This interface must open in order for nucleotides to leave, so nucleotide dissociation is greatly slowed. NBD dimerization requires a conformational change in the transmembrane domain of SUR1. This conformational change is communicated to Kir6.2 to increase the favorability of channel opening. (*bottom, left*) Sulfonylureas prevent the conformational change in the transmembrane domain of SUR1 and preclude NBD dimerization. Therefore, nucleotide dissociation is rapid, even in the presence of $Mg^{2+}$ and no increase in open probability results from Mg-nucleotide binding to SUR1. (*bottom, right*) Potassium channel openers stabilize the closed-dimer conformation of the NBDs. As a result, nucleotide dissociation is very slow and channel activation is enhanced.
DOI: https://doi.org/10.7554/eLife.41103.022

binding of MgTNP-ADP to each NBS2 of SUR1 is independent and favors pore opening by the same amount. The data in *Figure 5b* were fit with the following expressions for binding and activation:

$$\frac{bound}{total} = \frac{LE[ADP]K_A(E[ADP]K_A+1)^3 + [ADP]K_A([ADP]K_A+1)^3}{L(E[ADP]K_A+1)^4 + ([ADP]K_A+1)^4} \tag{6}$$

$$P_{open,norm} = \frac{\left(\frac{L(E[ADP]K_A+1)^4}{L(E[ADP]K_A+1)^4+([ADP]K_A+1)^4}\right) - \left(\frac{L}{L+1}\right)}{\left(\frac{LE^4}{LE^4+1} - \frac{L}{L+1}\right)} \tag{7}$$

Previous attempts to determine the stoichiometry of Mg-nucleotide activation of $K_{ATP}$, that is how many SUR1 subunits must bind nucleotide in order to activate current, have produced varying results (*Babenko, 2008*; *Hosy and Vivaudou, 2014*; *Tammaro et al., 2005*). We also fit MgTNP-ADP binding to NBS2 and current activation for SUR1-T1397* with models in which fixed stoichiometries of 1–4 nucleotide-bound SUR1 subunits were necessary to increase $P_{open}$ (*Figure 5—figure supplement 1*, *Figure 5—figure supplement 2*). All of these models used the same assumptions as

the MWC model in *Figure 5b*, but none fit the data as well. An inspection of our model fits (*Figure 5—figure supplement 2*) shows that the residuals for the MWC model and the model in which two bound nucleotide molecules activate $K_{ATP}$ are small and centered around zero. The other three models show systematic errors. Whereas the model in which two binding events opens $K_{ATP}$ provides a reasonable fit to the data, this model also predicts $E \approx 1$, that is no activation. The MWC fits to our data predicts $E = 2.2$, which is similar to the value of 1.76 that we previously estimated from electrophysiology data (*Proks et al., 2010*; *Vedovato et al., 2015*). This would favor the MWC model. However, given the scatter in the data, further experimentation will be required to truly distinguish between these competing models (see below).

Our binding and activation data were determined in separate experiments. Whereas the equations for binding and current activation both have a term ($L$) representing the intrinsic opening equilibrium, there is no way of knowing the $P_{open}$ of channels in an unroofed membrane. We expect $P_{open}$ to be quite low, as several minutes may elapse between cell unroofing and the completion of a given experiment, during which time channel rundown may occur (e.g. due to dissociation/degradation of $PIP_2$). Ideally, to derive more quantitative values, experiments should be performed using patch-clamp fluorometry in which binding and current are measured simultaneously (*Biskup et al., 2007*; *Zheng and Zagotta, 2000*). However, we were unable to obtain inside-out patches from cells expressing SUR1-T1397*/Kir6.2-G334D with sufficient fluorescence intensity to measure reliable spectra.

In our study, the concentration-response relationship for current activation was normalized to the minimum and maximum current values, as reflected in *Equation 7*. If the true maximum and minimum $P_{open}$ in a given experiment could be accurately determined, then the quantity $E$ could be computed directly from the minimum $P_{open}$ in the absence of ligand, $L/(L + 1)$, and the maximum $P_{open}$ at saturating nucleotide concentrations, $E^4L/(E^4L + 1)$, and our models would be further constrained. In practice, however, these values are difficult to measure for $K_{ATP}$ in the presence of $Mg^{2+}$ as channel rundown due to loss of $PIP_2$ results in an intrinsic $P_{open}$ ($L$) that is constantly changing (*Proks et al., 2016*). Nevertheless, our analysis provides useful qualitative information regarding the mechanism by which nucleotide binding to NBS2 of SUR1 activates Kir6.2.

The affinities we measured for TNP-ADP and TNP-ATP binding are higher than those previously estimated for ATP and ADP from photoaffinity labeling studies on SUR1 in the absence of Kir6.2 or from ATPase assays preformed on isolated NBD2 fusion proteins (*de Wet et al., 2007*; *Matsuo et al., 2000*). This may reflect the absence of Kir6.2 or a higher affinity of TNP nucleotides for NBS2 of SUR1 compared to ATP and ADP, as observed for other ABC transporters (*Oswald et al., 2008*). Nevertheless, the $EC_{50}$ values obtained from MgTNP-ATP and MgTNP-ADP binding to SUR1-Y1353*/Kir6.2 were in reasonably good agreement with those measured for MgATP and MgADP activation of SUR1/Kir6.2-G334D currents (*Proks et al., 2010*). It is unlikely that the introduction of ANAP modified the binding affinity of NBS2 as the apparent affinity for TNP-nucleotides in the absence of $Mg^{2+}$ was unaffected by the site at which ANAP was inserted (*Table 2*). The difference in apparent affinities between SUR1-Y1353* and SUR1-T1397* in the presence of $Mg^{2+}$ may indicate a difference in the ability of bound nucleotides to promote NBD dimerization.

MgADP can activate channels to an equal or greater extent than MgATP (*Proks et al., 2010*). Therefore, it is clear that the act of ATP hydrolysis *per se* is not required for channel activation, even though NBS2 is competent to hydrolyze ATP (*de Wet et al., 2007*; *Matsuo et al., 1999*). It is often stated that activation of $K_{ATP}$ channels by $Mg^{2+}$-ATP proceeds through hydrolysis of MgATP to MgADP (*Zingman et al., 2001*; *Zingman et al., 2002*). However, there is no evidence that an irreversible step like ATP hydrolysis occurs during the gating cycle of the $K_{ATP}$ channel in the presence of MgATP, as is the case for the closely related ABC protein CFTR (*Choi et al., 2008*; *Csanády et al., 2010*). Furthermore, high concentrations of ATP have been shown, in radioligand binding assays, to cause SUR1 to change conformation in the absence of $Mg^{2+}$, under which conditions ATP hydrolysis is not expected (*Ortiz et al., 2013*). $Mg^{2+}$-free ATP, at high concentrations, was also recently demonstrated to activate $K_{ATP}$ formed by SUR1-Q1179R, a gain-of-function SUR1 mutation, paired with Kir6.2-G334D, as well as SUR constructs in which the Walker$_B$ glutamate was mutated to glutamine (*Sikimic et al., 2018*). As lack of $Mg^{2+}$ or mutation of the catalytic Walker$_B$ motif do not support MgATP hydrolysis, this suggests that ATP can activate $K_{ATP}$ directly. No $Mg^{2+}$-free ATP activation was demonstrated when wild-type SUR1 was expressed with Kir6.2-G334D and

we see no indication in our data that nucleotides caused SUR1 to change conformation in the absence of $Mg^{2+}$ (*Figure 2*) (*Sikimic et al., 2018*).

Our data suggest that ATP hydrolysis did not occur over the course of our experiments, as the off-rate for MgTNP-ATP was faster than that of MgTNP-ADP ($t_{0.5}$ = 10.5 s vs 14.7 s). If MgTNP-ATP were hydrolysed to MgTNP-ADP before being released (as is thought to be true of many ABC family members, including CFTR) the dissociation rates should have been similar. The observation that MgTNP-ATP dissociation was faster than MgTNP-ADP dissociation may reflect a lack of MgTNP-ATP hydrolysis during the course of our experiments. Alternatively, this may indicate a destabilizing effect of inorganic phosphate on MgTNP-ADP binding to NBS2 (which would not be present when binding MgTNP-ADP alone, but would be present as a product of MgTNP-ATP hydrolysis) on binding.

Our ability to directly measure nucleotide binding in real time will be of considerable value for future investigations. This will allow us to address the role of nucleotide binding at NBS1 and Kir6.2, and explore the mechanistic ramifications of disease-associated $K_{ATP}$ mutations. Furthermore, simultaneous binding and current measurements will yield precise information regarding the coupling of stimulatory and inhibitory nucleotide binding to the channel pore. This coupling of ligand binding to an effector domain (the channel pore) that lies tens of Å away provides an excellent model for studying long-range communication within protein complexes. It has also not escaped our notice that our method is readily adaptable to the study of other ABC proteins, ATP-gated channels like P2X receptors (for which different subtypes demonstrate distinct selectivity for MgATP vs. ATP) and any protein for which there is a suitable fluorescent ligand (*Li et al., 2013*).

# Materials and methods

**Key resources table**

| Reagent type (species) or resource | Designation | Source or reference | Identifiers | Additional information |
|---|---|---|---|---|
| Cell line | HEK-293T | LGC Standards (ATCC CRL-3216) | | |
| Transfected construct (*Escherichia. coli*) | pANAP | Addgene | | |
| Transfected construct | pcDNA4/TO | Addgene | | |
| Transfected construct (*Aequorea victoria*) | pCGFP_EU | Gouaux Laboratory (Vollum Institute, Portland OR USA); | | |
| Transfected construct (*Homo sapiens*) | peRF1-E55D | Chin Laboratory (MRC Laboratory of Molecular Biology, Cambridge UK); | | |
| Antibody | Monoclonal ANTI-FLAG M2 antibody | Sigma-Aldrich | (Sigma-Aldrich Cat# F3165, RRID:AB_259529) | |
| Antibody | Anti-HA High Affinity; Rat monoclonal antibody (clone 3F10) | Roche | (Roche Cat# 11867423001, RRID:AB_10094468) | |
| Antibody | Sheep Anti-Mouse IgG ECL Antibody, HRP Conjugated | GE Healthcare | (GE Healthcare Cat# NA9310-1ml, RRID:AB_772193) | |
| Antibody | Peroxidase-AffiniPure Goat Anti-Rat IgG (H + L) antibody | Jackson Immuno Research Labs | (Jackson Immuno Research Labs Cat# 112-035-003, RRID:AB_2338128) | |
| Chemical compound, drug | trinitrophenyl-ATP (TNP-ATP) | Jena Bioscience (Jena, Germany) | | |

*Continued on next page*

*Continued*

| Reagent type (species) or resource | Designation | Source or reference | Identifiers | Additional information |
|---|---|---|---|---|
| Chemical compound, drug | trinitrophenyl-ADP (TNP-ADP) | Jena Bioscience (Jena, Germany) | | |
| Chemical compound, drug | L-3-(6-acetyl naphthalen-2-ylamino)—2-aminopropionic acid | Asis Chemicals (Waltham, MA) | | |

## Molecular biology

*Homo sapiens* SUR1 and Kir6.2 were subcloned into pcDNA4/TO (Invitrogen; Carlsbad, CA) for expression in HEK-293T cells. For experiments with GFP-labeled channels (*Figure 1—figure supplement 1b,c*), SUR1 was cloned into pCGFP_EU (*Kawate and Gouaux, 2006*). Mutagenesis, including insertion of amber (TAG) stop codons, was performed using the QuikChange kit (Agilent; Santa Clara, CA) or by subcloning synthetic oligonucleotides (Sigma; St. Louis, MO) between convenient restriction sites. All clones were verified by sequencing (DNA Sequencing and Services; Dundee, Scotland). pCDNA4/TO and pANAP were obtained from Addgene. peRF1-E55D (*Homo sapiens*) and pCGFP_EU (*Aequorea victoria*) were kind gifts from the Chin Laboratory (MRC Laboratory of Molecular Biology, Cambridge UK) and Gouaux Laboratory (Vollum Institute, Portland OR USA), respectively.

## Cell culture and expression of fluorescently tagged protein

HEK-293T cells were obtained from and verified/tested for mycoplasma by LGC Standards (ATTC CRL-3216; Middlesex, UK). Frozen stocks were prepared directly from these cells and stored in liquid nitrogen. Working stocks were routinely replenished from frozen following 15–25 passages. Our working stock tested negative for mycoplasma contamination using the MycoAlert Mycoplasma Detection Kit (Lonza Bioscience; Burton on Trent, UK). For experiments, cells were grown in 6-well plates (STARLAB; Milton Keynes, UK) or 30 mm culture dishes (STARLAB) on untreated 30 mm borosilicate coverslips (Thickness # 1, VWR International; Radnor, PA) or on poly-D-lysine coated FluoroDishes (FD35-PDL-100, World Precision Instruments; Hitchin, UK) in Dulbecco's Modified Eagle Medium (Sigma) supplemented with 10% fetal bovine serum, 100 U/mL penicillin, and 100 µg/mL streptomycin (Thermo Fisher Scientific; Waltham, MA) at 37°C, 5%/95% $CO_2$/air.

SUR1 constructs were site-specifically tagged with the fluorescent amino acid ANAP using amber codon suppression as described by Chatterjee *et al.* (*Chatterjee et al., 2013*) 30 mm dishes of HEK-293T cells were co-transfected with pANAP (0.5–1 µg), Kir6.2 (0.5 µg), and SUR1 (1–1.5 µg) constructs containing an amber stop codon (TAG) at amino acid position 1353 or 1397 (1353[stop] or 1397[stop]) using TransIT transfection reagent (Mirus Bio LLC; Madison, WI) in a ratio of 3 µL of TransIT per µg of total DNA. Following transfection the media was supplemented with 20 µM ANAP (free acid or methyl ester). pANAP encodes a tRNA/tRNA synthetase pair specific for ANAP. In the presence of ANAP, cells transfected with pANAP generate tRNAs charged with ANAP that recognize the amber stop codon. Thus, full-length ANAP-tagged SUR can be produced. Proper co-assembly of Kir6.2 subunits with SUR1 subunits is ensured by the fact that the two subunits rely on one another for proper exit from the ER and trafficking to the plasma membrane (*Zerangue et al., 1999*). An exception to this may be SUR1_GFP, which has been shown to traffic independently of Kir6.2 (*Makhina and Nichols, 1998*). After transfection, cells were cultured at 33° C, 5%/95% $CO_2$/air to slow growth and increase the per-cell protein yield (*Lin et al., 2015*). To enhance expression of full-length, ANAP-labeled proteins, 1–2 µg of a plasmid containing a dominant negative eukaryotic release factor 1 (peRF1-E55D) was included in the transfection mix as indicated (*Schmied et al., 2014*). Experiments were performed 2–5 days post transfection.

## Preparation of unroofed membrane fragments for imaging

Unroofed membranes were prepared using a modification of a protocol from the Heuser laboratory (*Heuser, 2000*; *Usukura et al., 2012*; *Zagotta et al., 2016*). Briefly, a coverslip containing adherent

cells was removed from the cell culture medium and a small piece (0.5–1 cm) was broken off using jewelers forceps (#5) and washed in phosphate buffered saline (PBS, Thermo Fisher Scientific) diluted 1:3 in deionized water. Cells were exposed 3 × 10 s to a solution of 0.01% poly-L-lysine (Sigma) with intervening washes in diluted PBS to adhere them more firmly to the coverslips. The coverslip fragment was then placed in a 30 mm culture dish filled with 2–3 mL of 1/3X PBS and sonicated very briefly (~100 ms) using a probe sonicator (Vibra-cell; Newtown, CT), leaving behind adherent plasma membrane and sometimes additional cellular material. The unroofed fragments in our experiments were sometimes visible under bright-field illumination, a potential result of our modified 'unroofing' procedure. Morphologically, such cell fragments resemble the 'partially unroofed' cells identified by Usukura *et al.* (*Usukura et al., 2012*) Cells cultured on poly-D-lysine coated FluoroDishes were unroofed in diluted PBS with no additional poly-L-lysine treatment. This procedure yielded a higher percentage of completely unroofed plasma membranes (i.e. nearly invisible with brightfield illumination).

## Microscopy/spectroscopy

Unroofed membrane fragments were imaged directly on FluoroDishes (FD35-PDL-100, World Precision Instruments) or on broken coverslips placed in a FluoroDish (FD3510, World Precision instruments) using a Nikon TE2000-U microscope equipped with a 40x (S Fluor, 1.3 NA; Nikon, Kingston Upon Thames, UK) or 100x (Apo TIRF, 1.49 NA, Nikon) oil-immersion objective and low-fluorescence immersion oil (MOIL-30, Olympus; Southend-on-Sea, UK) or a 60x water immersion objective (Plan Apo VC, 1.20 NA, Nikon). ANAP was excited using a ThorLabs LED source (LED4D067) at 385 nm connected to the microscope using a liquid light guide in series with a 390/18 nm band-pass filter (MF390-18, ThorLabs; Newton, NJ) and MD416 dichroic (ThorLabs). For imaging, emitted light was filtered through a 479/40 nm band-pass filter (MF479-40, ThorLabs). GFP-tagged constructs were imaged using a similar setup with excitation at 490 nm, a 480/40 band-pass excitation filter (Chroma; Bellows Falls, VT), a DM505 dichroic mirror (Chroma), and 510 nm long-pass emission filter (Chroma). All images were acquired using a PIXIS 400B CCD camera (Princeton Instruments; Trenton, NJ). For spectroscopy, ANAP was excited as above, but emitted light was passed through a 400 nm long-pass filter (FEL0400, ThorLabs) and directed through the slit of an Isoplane 160 spectrometer (300 g/mm grating; Princeton Instruments) in series with the camera. The acquired images retained spatial information in the y-dimension, with the x-dimension replaced by wavelength (*Figure 1—figure supplement 1a*). Exposure times were typically 100 ms for brightfield images, 1–10 s for steady-state spectra and 1–5 s for time-course experiments. The emission spectrum of ANAP-labeled SUR1 showed a peak around 470 nm, which was used to differentiate labeled channels from autofluorescence (which typically had a broad emission spectrum), from cytoplasmic ANAP (or potentially ANAP-conjugated to tRNA), which peaks at ~480 nm or from bright, fluorescent debris on the coverslip which was morphologically distinct from unroofed membranes and had spectra peaking at either ~450 nm or 480–485 nm.

To verify that the fluorescence signal we observed in unroofed membrane fragments derived from ANAP-labeled SUR1, we co-transfected cells with Kir6.2 and GFP-tagged SUR1 with or without an amber stop codon at position Y1353. After unroofing, membranes from transfected cells could be identified by the green fluorescence of SUR1_GFP/Kir6.2. *Figure 1—figure supplement 1b* shows that membranes expressing SUR1_GFP-Y1353*/Kir6.2 were brightly fluorescent for both GFP and ANAP, whereas those from SUR1_GFP expressing cells had only GFP fluorescence, even though ANAP was included in the cell-culture medium. We also acquired emission spectra from membranes expressing SUR1_GFP-Y1353*/Kir6.2 (*Figure 1—figure supplement 1c*). The intensity of the ANAP peak in our fluorescence was linearly proportional to the peak GFP fluorescence, and the intensity of the ANAP peak extrapolated to zero GFP (i.e. no expression of SUR1_GFP-Y1353*) was zero. Taken together, these data suggest that non-specific ANAP background was very low. We also performed western blotting on FLAG-tagged SUR1-Y1353[stop] constructs to verify that we produced full-length, ANAP-labeled SUR1, and that cells were unable to read through the amber stop codon to produce full-length protein in the absence of ANAP (*Figure 1—figure supplement 1d*).

To prevent accumulation of fluorescent nucleotides in the bath, FluoroDishes were perfused with either (in mM) 140 KCl, 20 NaCl, 10 MgCl$_2$, 10 HEPES pH 7.2 with NaOH or 140 KCl, 20 NaCl, 1 EDTA, 10 HEPES pH 7.2 with NaOH using a peristaltic pump (Ismatec; Wertheim, Germany). Unroofed membranes were directly perfused with nucleotides using an 8-channel μFlow perfusion

system (ALA Scientific Instruments; Farmingdale, NY). Solution change with the µFlow was rapid (τ = 0.73 s±0.02 s, n=5) as measured from single exponential fits to wash-out of 50 µM tetramethylrhodamine-5-maleimide (TMRM) after switching to a solution with no TMRM. TMRM was excited with a 565 nm LED with a D540/25X band-pass filter (Chroma) and DM565 dichroic mirror (Chroma). Emitted light was collected through a D605/55M band-pass filter (Chroma). Solution exchange was also monitored during experiments with TNP-nucleotides by examining the emission spectrum of a background region in each image. However, the detection limit for TNP-nucleotides in solution was low as unbound nucleotides have a low quantum yield (enhanced by protein binding) and are not excited via FRET, as is the case for nucleotides bound to ANAP-labeled SUR1 (*Broglie and Takahashi, 1983*).

Images and spectra were acquired using LightField software (Princeton Instruments). The solution changer, camera, and light source were all controlled using pClamp 10.5 (Molecular Devices; San Jose, CA) and a DigiData 1440 A/D converter. All data were acquired at room temperature (18°−22° C).

## Analysis of images and spectra

Images were adjusted and displayed using Fiji (*Schindelin et al., 2012*). Spectra were analyzed using custom code written in Matlab (Mathworks; Natick, MA). The code was deposited on GitHub (https://github.com/mpuljung/spectra-analysis; copy archived at https://github.com/elifesciences-publications/spectra-analysis) (*Puljung, 2019a*; *Puljung, 2019b*). Regions of interest corresponding to fluorescent membrane fragments were manually selected from the spectral images and averaged for each wavelength. A background region of similar size was selected and the average background spectrum was subtracted from the spectrum of the region of interest. The peak fluorescence (~470 nm) was automatically selected by determining the maximum of a boxcar average of 31 points (corresponding to ~10 nm) of the data in the absence of nucleotides. Data were corrected for photobleaching by fitting an exponential decay ($F = Ae^{t/\tau} + (1- A)$) to the normalized peak of 5–6 images acquired prior to washing on TNP-nucleotides and dividing the raw spectra by the resulting fit at each cumulative exposure time (*Figure 1—figure supplement 5*). Photobleaching rates were similar regardless of the position of ANAP in SUR1. For concentration-response relationships, corrected peak data (at ~470 nm) were normalized by the ANAP fluorescence at zero [TNP-nucleotide] ($F/F_{max}$) and displayed as $1-F/F_{max}$. On occasion data were normalized instead to the fluorescence at 50 nM nucleotide, as we consistently saw no effect at this concentration. TNP-nucleotide concentration-response relationships were fit with the Hill equation

$$1 - \frac{F}{F_{max}} = E_{max} * \frac{[TNP]^h}{EC_{50}^h + [TNP]^h} \qquad (1)$$

where $E_{max}$ is the FRET efficiency at saturating concentrations, $[TNP]$ is the concentration of TNP-nucleotide, $EC_{50}$ is the half-maximal concentration and $h$ is the Hill slope.

To generate the plot of GFP fluorescence vs. ANAP fluorescence in *Figure 1—figure supplement 1c*, spectra of ANAP- and GFP-tagged SUR1 subunits expressed in unroofed membrane fragments were acquired as above. We were able to resolve a GFP peak at 510 nm with 385 nm excitation, even though this wavelength is far from the $\lambda_{max}$ for GFP. It is possible that GFP emission was enhanced via FRET between ANAP and GFP. The ANAP peak fluorescence at 470 nm did not overlap with the GFP emission peak. However, the peak GFP at 510 nm was contaminated by the shoulder of the ANAP peak. Therefore, the GFP fluorescence was corrected by subtracting the averaged (from 16 membranes) fluorescence of ANAP incorporated into SUR1, scaled to match the 470 nm peak in the ANAP-GFP spectrum.

To measure the time course of nucleotide unbinding, the fluorescence at the TNP-nucleotide peak (usually around 560 nm) was plotted as a function of time after exchanging to a zero-nucleotide solution, as following the time course of the rise in ANAP fluorescence was complicated by photobleaching. After complete wash out of nucleotides, the fluorescence at 560 nm was non-zero, as the shoulder of the ANAP emission spectrum extends into the range of wavelengths at which TNP-nucleotides emit. Therefore, the time course data were corrected by subtracting a linear fit to the data following complete washout. This corrected for the shoulder of the ANAP spectrum as well as

the continued photobleaching of ANAP. Data were normalized to the maximum fluorescence at t = 0 and fit with single-exponential decays

$$\frac{F}{F_{max}} = e^{-t/\tau} \qquad (2)$$

or double-exponential decays

$$\frac{F}{F_{max}} = A_1 e^{-t/\tau_1} + A_2 e^{-t/\tau_2}. \qquad (3)$$

Wash-out time courses following exchange to zero TNP-nucleotide solution should be independent of the nucleotide concentration before wash. The majority of our time-course experiments were conducted at the conclusion of our concentration-response experiments, and therefore reflect wash after applying a concentration of 1 mM. For experiments in diazoxide and tolbutamide, the nucleotide concentration prior to wash was 50 µM.

In some experiments, there was evidence of cross-contamination of our wash solution or low-nucleotide solutions via back-flow of solutions with higher concentrations of nucleotide (as evident by accumulation of yellow TNP-nucleotides). We excluded data from such experiments and maintained/replaced valves on the µFlow as needed.

## Electrophysiology

Currents were recorded from inside-out membrane patches excised from transfected HEK-293T cells. Pipettes were pulled to a resistance of 1–5 MΩ and filled with an extracellular solution containing (in mM) 140 KCl, 1.2 MgCl₂, 2.6 CaCl₂, and 10 HEPES, pH 7.4 or 140 KCl, 1 EGTA, 10 HEPES, pH 7.3. The intracellular (bath) solutions contained either (in mM) 107 KCl, 2 MgCl₂, 1 CaCl₂, 10 EGTA, and 10 HEPES, pH 7.2 for experiments showing nucleotide-dependent activation or 140 KCl, 1 EDTA, 1 EGTA, 10 HEPES pH 7.3 for experiments showing nucleotide-dependent inhibition. Nucleotides were added as indicated. Patches were perfused with an 8-channel µFlow perfusion system. Data were acquired at a holding potential of −60 mV using an Axopatch 200B amplifier and Digidata 1322A digitizer with pClamp 9.0 software (Molecular Devices). Currents were digitized at 10 kHz and low-pass filtered at 1 kHz. $K_{ATP}$ channels run down in excised patches (reviewed by Proks et al.) (*Proks et al., 2016*). For inhibitory concentration-response relationships, rundown was corrected by alternating test nucleotide solutions with nucleotide-free (control) solutions and expressing the test currents as a fraction of the average of the control currents before and after the test solution. Corrected data were fit to the following expression:

$$\frac{I}{I_{max}} = 1 - \frac{[TNP]^h}{IC_{50}^h + [TNP]^h}. \qquad (4)$$

with $IC_{50}$ representing the half-maximal inhibitory concentration. For activation concentration-responses, the same protocol was used, but test concentrations were alternated with a saturating concentration (1 mM) of MgADP (*Proks et al., 2010*). Data were displayed at the current magnitude in the test concentration minus the current in zero nucleotide ($I$-$I_{min}$) divided by the current in saturating nucleotide (1 mM MgADP) minus the current in zero nucleotide ($I_{max} - I_{min}$) and fit to the following expression:

$$\frac{I - I_{min}}{I_{max} - I_{min}} = \frac{[TNP]^h}{EC_{50}^h + [TNP]^h} \qquad (5)$$

Currents were leak corrected by subtracting the remaining current after complete rundown.

## Western blotting

In order to detect SUR1 via western blot, a triple FLAG-tagged construct (3xFLAG_SUR1, MDYKDHDGDYKDHDIDYKDDDDK; tag in an extracellular loop between amino acid positions T1042 and L1043) was used with or without amber stop codons at amino acid positions 1353 or 1397. HEK-293T cells were transfected in 6-well plates using the conditions described above, but with 1 µg of 3xFLAG_SUR1 DNA. After 48 hr, cells were washed and harvested with gentle

pipetting in 1 mL of PBS (Sigma). Cells were pelleted (4 min at 200 x $g$ yielding 20–50 mg pellets) and solubilized in 50–100 µL of 0.5% Triton X-100 in 110 mM potassium acetate, buffered at pH 7.4. 125 U of benzonase (Sigma) was added to each sample and samples were incubated at room temperature for 20 min. 7.5 µL of each sample was mixed with 2.5 µL of NuPAGE LDS sample buffer and 1 µL NuPAGE sample reducing agent (Invitrogen) and run on a NuPAGE 4–12% Bis-Tris gel (Invitrogen) at 200 V for 40 min in 1X MOPS-SDS buffer (Invitrogen). Protein was transferred overnight at 10 V to an Immobilon-P membrane (Merck Millipore; Burlington, MA) in 25 mM Tris, 192 mM glycine, and 20% methanol +0.1% SDS. Following transfer, the membrane was rinsed in TBS-Tw (150 mM NaCl, 25 mM Tris, pH 7.2, and 0.05% Tween 20) and incubated (with shaking) for 30 min in TBS-Tw +5% milk. The membrane was rinsed three times in TBS-Tw and incubated for 30 min with primary antibody (M2 anti-FLAG from Sigma, diluted 1:500 in TBS-Tw). The membrane was rinsed three more times in TBS-Tw and incubated for 30 min with secondary antibody (HRP-conjugated sheep anti-mouse IgG, GE Healthcare Life Sciences; Freiburg, Germany) diluted 1:20,000 in TBS-Tw +1% milk. Finally, the membrane was rinsed 3 × 10 min in TBS-Tw and developed using SuperSignal West Femto Max Sensitivity Substrate (Thermo Fisher Scientific). Images were collected using a C-DiGit scanner (Licor Biosciences; Lincoln NE) and band intensities were quantified using custom code written in MATLAB (https://github.com/mpuljung/spectra-analysis) (*Puljung, 2019c*).

Western blotting was performed (*Figure 1—figure supplement 1d*) on the total protein from HEK-293T cells transfected with Kir6.2 and 3xFLAG-tagged SUR-Y1353$^{stop}$ DNA under various conditions. As the 3xFLAG tag was N-terminal to any introduced stop codons in SUR1, we used it to probe for both full-length and truncated SUR1. As a positive control, we transfected 3x-FLAG-tagged SUR1 with no stop codon, which was only expected to produce full-length protein (lane 1). This protein ran above our 140 kDa marker as expected (predicted MW is ~180 kDa). In the absence of ANAP or in cells transfected without the pANAP plasmid, only truncated 3xFLAG_SUR1 protein was evident (lanes 4 and 5), confirming that there was no full-length protein produced by accidental read-through of the amber (TAG) codon. In the presence of both pANAP and ANAP, we obtained a mixture of full-length (66%) and truncated (34%) SUR1. The majority of our experiments in unroofed membrane fragments were performed under such conditions, as the signals we measured (fluorescence) were derived solely from full-length (i.e. ANAP-labeled) channels. Finally, we were able to increase expression of full-length SUR1 as needed by transfecting an additional plasmid, peRF1-E55D, which encodes a dominant negative ribosomal release factor (*Schmied et al., 2014*). With the additional expression of ERF1-E55D, we obtained 92% full-length protein (*Figure 1—figure supplement 1d*). A similar approach was taken with SUR1-T1397$^{stop}$ (*Figure 4—figure supplement 1d*).

In parallel experiments, we expressed SUR1_GFP-Y1353$^{stop}$ in the absence of ANAP. Under such conditions, we observed a few cells with diffuse GFP fluorescence throughout the cytoplasm. We believe that the signal from such cells was derived from soluble GFP generated from an internal methionine after position 1353 and not read-through of the amber stop codon, as the distribution of this protein differed from that of SUR1_GFP (which was confined to the plasma membrane and presumably ER/Golgi). We never observed GFP fluorescence in unroofed membranes of such cells.

## Surface expression assay

Surface expression of SUR1 constructs was assayed by their ability to chaperone HA-tagged Kir6.2 subunits to the plasma membrane using an adaptation of the method of Zerangue *et al.* (*Zerangue et al., 1999*) The HA tag plus linker (YAYMEKGITDLAYPYDVPDY) was inserted in the extracellular region following helix M1 of Kir6.2 between amino acids L100 and A101. HEK-293T cells were cultured in 12-well plates (STARLAB) on 19 mm cover slips treated with poly-L-lysine. Transfections were performed as described above, but DNA amounts were adjusted for the smaller scale (0.08 µg HA-tagged Kir6.2, 0.14 µg SUR1 constructs, 0.14 µg pANAP, 0.14 µg peRF1-E55D). Cells were incubated in DMEM (+10% FBS, 100 U/mL penicillin, and 100 µg/mL streptomycin) at 33°C in a 95% air/5% $CO_2$ atmosphere for 48 hr in the presence or absence (as indicated) of 20 µM ANAP. Following incubation, cells were rinsed on ice with PBS and fixed for 30 min in PBS + 10% neutral buffered formalin. Cells were subsequently washed twice in PBS and blocked with 1% bovine serum albumin (BSA) in PBS for 30 min at 4° C. Cells were then incubated for 1 hr at 4° C in PBS + 1% BSA with a 1:1000 dilution of rat anti-HA monoclonal antibody (Roche; Basel, Switzerland) and washed 5 × 10 min on ice with PBS + 1% BSA. HRP-conjugated goat anti-rat polyclonal

antibody (diluted 1:2000, Jackson ImmunoResearch; Ely, UK) was applied in PBS + 1% BSA and cells were incubated an additional 30 min at 4° C. Finally, cells were washed 4 × 10 min in PBS + 1% BSA and 5 × 10 min in PBS on ice. After the wash, coverslips were removed from PBS and placed in clean, untreated 35 mm culture dishes. 300 µL of SuperSignal ELISA Femto Maximum Sensitivity Substrate (Thermo Fisher Scientific) were added and the luminescence was measured using a Glomax 20/20 Luminometer (Promega; Madison, WI) after a 10 s incubation. For display purposes, data were normalized to the mean of the luminescence with wild-type SUR1. Background was assessed with cells that were transfected with HA-tagged Kir6.2, but no SUR1.

## Chemicals

ANAP trifluoroacetate salt and methyl ester were obtained from Asis Chemicals (Waltham, MA). A 1 mM stock of the salt was prepared in 30 mM NaOH and stored at −80° C. The methyl ester was dissolved at 5 mM in DMSO and stored at −20° C. The two forms were used interchangeably, as the final product (when ANAP is incorporated into a protein) is identical and in our experience both free acid and methyl ester were membrane permeant. Trinitrophenyl (TNP) nucleotide analogues were purchased from Jena Bioscience (Jena, Germany). 1 mM stocks were prepared in buffer and stored at −20° C. Stocks were verified using UV-Vis spectroscopy (Beckman Coulter DU800 spectrophotometer; Pasadena, CA) as needed. Paper chromatography (mobile phase 40:10:25 $n$-butanol, glacial acetic acid: $H_2O$) was used to verify that older TNP-ATP stocks had not undergone hydrolysis. All other chemicals were from Sigma. Diazoxide was prepared in a 34 mM stock in 100 mM KOH and stored at −20° C before use. Tolbutamide was dissolved at 100 mM in DMSO and stored at −20° C.

## Data presentation and statistics

Error bars represent ±SEM. The boxes in *Figure 2e* and *Figure 4—figure supplement 1a,b* represent ±SD. The number of experiments (n) represents the number of patches or membranes used in a given experiment. These can be found in the tables for binding, electrophysiology, and time course data, and figure legends for other experiments. For surface expression assays, n represents the number of transfected coverslips. Plots and curve fitting were generated using Origin 9.1 (OriginLab Corporation; Northampton, MA). Curves in the figures (and parameters reported in the figure legends) represent fits to the combined data sets from multiple experiments, with errors representing the standard error of the fits. Shaded regions indicate 95% confidence intervals. Fit parameters reported in Tables are the mean ±SEM (*Tables 1*, *2* and *5*) or ±SD (*Tables 3* and *4*) of the fits to individual experiments. Pairwise comparisons were performed using two-tailed Student's $t$-tests with the Welch correction (not assuming equal variance) and the Bonferroni correction for multiple comparisons, as needed. Comparisons of $EC_{50}$ values were performed on log-transformed data as the untransformed $EC_{50}$ is not normally distributed. Protein structures were displayed using PyMOL (Schrödinger, LLC; New York, NY).

## Acknowledgments

We wish to thank Idoia Portillo and Raul Terron Exposito for excellent technical assistance and Gregor Sachse, Chris Miller, and David Gadsby for helpful discussions. James Cantley provided access to the western blot scanner. This work was supported by the European Research Council (grant 322620), the Biotechnology and Biological Sciences Research Council (BB/R002517/1), the John Fell Fund, and the Wellcome Trust Oxion graduate program.

## Additional information

### Funding

| Funder | Grant reference number | Author |
| --- | --- | --- |
| Biotechnology and Biological Sciences Research Council | BB/R002517/1 | Michael Puljung Frances Ashcroft |
| H2020 European Research Council | 322620 | Michael Puljung Natascia Vedovato Frances Ashcroft |

| Wellcome Trust | Oxion Graduate Program | Samuel Usher |
| John Fell Fund, University of Oxford | | Michael Puljung |

The funders had no role in study design, data collection and interpretation, or the decision to submit the work for publication.

## Author contributions

Michael Puljung, Conceptualization, Resources, Data curation, Software, Formal analysis, Supervision, Funding acquisition, Validation, Investigation, Visualization, Methodology, Writing—original draft, Project administration, Writing—review and editing; Natascia Vedovato, Data curation, Formal analysis, Investigation, Writing—review and editing; Samuel Usher, Data curation, Funding acquisition, Investigation; Frances Ashcroft, Conceptualization, Resources, Supervision, Funding acquisition, Project administration, Writing—review and editing

## Author ORCIDs

Michael Puljung (iD) http://orcid.org/0000-0002-9335-0936
Samuel Usher (iD) https://orcid.org/0000-0002-2487-6547
Frances Ashcroft (iD) https://orcid.org/0000-0002-6970-1767

## Decision letter and Author response

Decision letter https://doi.org/10.7554/eLife.41103.027
Author response https://doi.org/10.7554/eLife.41103.028

# Additional files

## Supplementary files

• Transparent reporting form
DOI: https://doi.org/10.7554/eLife.41103.023

## Data availability

Data available from the Dryad Digital Repository: https://doi.org/10.5061/dryad.6mh0sv3

The following dataset was generated:

| Author(s) | Year | Dataset title | Dataset URL | Database and Identifier |
|---|---|---|---|---|
| Michael Puljung | 2019 | Data from: Activation mechanism of ATP-sensitive K+ channels explored with real-time nucleotide binding–Revised | https://doi.org/10.5061/dryad.6mh0sv3 | Dryad Digital Repository, 10.5061/dryad.6mh0sv3 |

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
