## [Decision Letter]

Thank you for submitting your article "Activation mechanism of ATP-sensitive K^+^ channels explored with real-time nucleotide binding" for consideration by *eLife*. Your article has been reviewed by three peer reviewers, including Kenton Swartz as the Reviewing Editor and Reviewer #1, and the evaluation has also been overseen by John Kuriyan as the Senior Editor. The following individuals involved in review of your submission have agreed to reveal their identity: Show-Ling Shyng (Reviewer #2); Marcel P. Goldschen-Ohm (Reviewer #3).

The reviewers have discussed the reviews with one another and the Reviewing Editor has drafted this decision to help you prepare a revised submission.

Summary:

The authors use FRET between a fluorescent unnatural amino acid ANAP and fluorescent analogs of ATP and ADP (TNP-ATP/ADP) to measure nucleotide binding to the SUR1 subunit of the K_ATP_ channel complex in unroofed HEK cells. They show that TNP-ADP/ATP bind both in the presence and absence of Mg, but that Mg slows the rate of unbinding. Unbinding is also either sped or slowed by tolbutamide or diazoxide, respectively, suggesting that Mg promotes a conformational change associated with channel gating, such as dimerization of the nucleotide binding domains. Based on this, the authors propose that examining the effects of perturbations (e.g. mutation) on binding both with and without Mg can be used to dissect its effects on initial binding steps vs. subsequent conformational changes, an idea which they apply to the mutation K1394A. The approach seems quite strong in that it provides direct measures of binding from channels in native membrane with various activation states promoted by either Mg or drugs. The data appear to be of good quality, and the dependence of the unbinding rate on Mg and both inhibitory and stimulatory ligands shows nicely that the stability of the bound state is coupled to states associated with channel activation. However, we are not completely sold on the idea that binding vs. conformational change can be easily separated solely on Mg-dependence as the authors suggest, and we think this issue requires more consideration in the manuscript. Also, there seem to be some issues with the kinetic modeling that need to be revised.

Essential revisions:

1) The Mg and drug-dependence of the unbinding rate clearly suggests that bound stability reflects activation state. However, across constructs (Y1353*, T1397* and Y1353* K2A), Mg right-shifts binding curves except in one case where it did not shift, suggesting that it reduces overall affinity. If it does slow the unbinding rate as the data show, then it must also slow the binding rate even more, even if it is also coupled to a conformational change. Thus, it does not seem possible to correlate a right shift of the binding curve with reduced stability of the bound state, and it seems that Mg must have a role in binding as well, and not solely act to promote dimerization after state-independent binding with or without Mg, which one would naively expect to only left-shift binding curves by trapping bound nucleotide. It would be good to openly discuss these issues. Also, for the K2A mutant, we would like to see the unbinding data in the absence of Mg as well, as assuming it will be unchanged based on the absence of a shift in the binding curve seems dangerous given the other results presented.

2) The fact that Y1353* containing channels could not be activated by MgTNP-ADP raises the concern that results with this construct may reflect a nonfunctional channel complex. Given the similar binding curves for T1397*, which can be activated, we are not averse to the authors suggestion that other results should also be similar. However, it would be comforting to see that the unbinding rate was also similarly affected by Mg and drugs in a functional channel.

3) The authors offered a couple of possible explanations for why Y1353* channels are not activated by Mg-nucleotides. The first is that Y1353* may disrupt functional coupling between NBD-dimerized SUR1 and Kir6.2 based on the quatrefoil structure and previous studies showing that mutation of an adjacent residue R1352 disrupts channel activation. Have the authors tried mutating Y1353 and see whether Mg-nucleotide activation is impaired? The second explanation is incorporation of the truncated SUR1 into the channel complex. However, judging from the western blot in Figure 1—figure supplement 1, the truncated protein is a minor species so if this were the mechanism then the truncated protein must have a strong dominant negative effect, which may have implications for functional stoichiometry of Mg-nucleotide activation. One easy way to clarify this issue is to co-express WT SUR1 and SUR1 truncated at Y1353 to see whether the resulting channels are functional.

4) There are two technical point that it would be good to explicitly address in the manuscript. What were the estimated bleaching time constants for ANAP in the various experimental conditions? Does MgTNP-ATP undergo hydrolysis during the course of the FRET experiment, and could this affect the results?

5) Since only NBS2 binding is monitored by FRET, the data does not inform the status of the NBS1 and whether or how NBS2 nucleotide binding is affected by NBS1 binding. This is important since the authors interpret the increased lifetime for TNP-ADP/ATP by Mg^2+^ as a result of conformational change i.e. NBD dimerization. Is it known whether the efficiency of dimerization changes depending on which nucleotide species is bound at NBS1? It would be good to have more discussion on previous studies about whether free or Mg-bound nucleotides bind to both NBS1 and 2, and whether activation can occur in the absence of Mg.

6) Concerning Figure 5—figure supplement 1, there are some issues with the modeling that may be due to E not being constrained to E>1 as suggested in the text (Discussion section). For example, given E=1 as reported for Figure 5—figure supplement 1B, pore opening should be independent of nucleotide binding, but that is not what is shown. Also, in Figure 5—figure supplement 1C, E=4.9e-14 implies the pore should be nearly fully closed after binding two or more nucleotides, which is clearly not what is depicted in the plot. Regardless, given the scatter in the data. we are dubious as to whether any but perhaps the most extreme model (i.e. panel e) can be ruled out, and perhaps focusing on something like this rather than arguing for a particular model would be an approach worth considering. Despite differences in RSS, model fits for, say panels a and d, are similar enough that it seems likely that they would have been considered to be not different elsewhere in the manuscript.

---

## [Author Response]

Essential revisions:1) The Mg and drug-dependence of the unbinding rate clearly suggests that bound stability reflects activation state. However, across constructs (Y1353*, T1397* and Y1353* K2A), Mg right-shifts binding curves except in one case where it did not shift, suggesting that it reduces overall affinity. If it does slow the unbinding rate as the data show, then it must also slow the binding rate even more, even if it is also coupled to a conformational change. Thus, it does not seem possible to correlate a right shift of the binding curve with reduced stability of the bound state, and it seems that Mg must have a role in binding as well, and not solely act to promote dimerization after state-independent binding with or without Mg, which one would naively expect to only left-shift binding curves by trapping bound nucleotide. It would be good to openly discuss these issues. Also, for the K2A mutant, we would like to see the unbinding data in the absence of Mg as well, as assuming it will be unchanged based on the absence of a shift in the binding curve seems dangerous given the other results presented.

Thank you for identifying this error in our modelling. We have modified our discussion in the Results section to state that Mg^2+^ must have a direct effect on nucleotide binding as well, and that the effect of the K2A mutation may in part be due to an effect on binding in the presence of Mg^2+^. We have also included more data in Figure 3F to show the dissociation rate of MgADP from K2A channels in the absence of Mg^2+^.

2) The fact that Y1353* containing channels could not be activated by MgTNP-ADP raises the concern that results with this construct may reflect a nonfunctional channel complex. Given the similar binding curves for T1397*, which can be activated, we are not averse to the authors suggestion that other results should also be similar. However, it would be comforting to see that the unbinding rate was also similarly affected by Mg and drugs in a functional channel.

We agree that these experiments are important and have made several attempts to perform them. We repeatedly observed residual nucleotide binding to SUR1T1397*/Kir6.2 after extensive washing (>4 min, see Author response image 1). It is possible that this may reflect a bona fide difference between T1397* and Y1353*. However, we have also been experiencing problems with our perfusion setup such that it has been difficult to remove nucleotides from the bath in a timely fashion. Even constructs for which we previously obtained complete nucleotide washout show residual binding in some experiments. As such, we are not confident of this result and prefer to leave it out of the final manuscript.

**Author response image 1. respfig1:** *Left*. Spectra acquired before (First) and after (Last) 4 min of washout. A residual nucleotide peak (530 nm) remains after washing. *Right*. Spectra taken from a background region. The nucleotide signal is smaller (there is no FRET to unbound nucleotide and the quantum efficiency for unbound TNP-nucleotides is low) but fails to completely wash out in 4 min. Note that the spectrum of the background region also contains autofluorescence from the culture dish.

3) The authors offered a couple of possible explanations for why Y1353* channels are not activated by Mg-nucleotides. The first is that Y1353* may disrupt functional coupling between NBD-dimerized SUR1 and Kir6.2 based on the quatrefoil structure and previous studies showing that mutation of an adjacent residue R1352 disrupts channel activation. Have the authors tried mutating Y1353 and see whether Mg-nucleotide activation is impaired? The second explanation is incorporation of the truncated SUR1 into the channel complex. However, judging from the western blot in Figure 1—figure supplement 1, the truncated protein is a minor species so if this were the mechanism then the truncated protein must have a strong dominant negative effect, which may have implications for functional stoichiometry of Mg-nucleotide activation. One easy way to clarify this issue is to co-express WT SUR1 and SUR1 truncated at Y1353 to see whether the resulting channels are functional.

We have expanded Figure 4—figure supplement 1 and added additional text to subsection “Correlation of binding with gating” to address this issue. We now show a comparison of full-length SUR/Kir6.2 channels with truncated channels, as well as channels in which we expect a mix of full-length and truncated subunits (SUR1-Y1353^stop^ expressed with pANAP), and channels in which nearly all the subunits should be full-length, ANAP-tagged channels. We have also included additional current traces showing a lack of activation of SUR1-Y1353*/Kir6.2-G334D. Based on this, we now favor the hypothesis that full-length Y1353* channels do not support activation.

4) There are two technical point that it would be good to explicitly address in the manuscript. What were the estimated bleaching time constants for ANAP in the various experimental conditions? Does MgTNP-ATP undergo hydrolysis during the course of the FRET experiment and could this affect the results?

We have included an additional figure (Figure 1—figure supplement 5) showing our process for correcting the data for photobleaching and comparing photobleaching rates between constructs. We have also included additional discussion in the Discussion section regarding ATP hydrolysis.

5) Since only NBS2 binding is monitored by FRET, the data does not inform the status of the NBS1 and whether or how NBS2 nucleotide binding is affected by NBS1 binding. This is important since the authors interpret the increased lifetime for TNP-ADP/ATP by Mg^2+^ as a result of conformational change i.e. NBD dimerization. Is it known whether the efficiency of dimerization changes depending on which nucleotide species is bound at NBS1? It would be good to have more discussion on previous studies about whether free or Mg-bound nucleotides bind to both NBS1 and 2, and whether activation can occur in the absence of Mg.

To our knowledge, it is not known how different species binding to NBS1 affect dimerization, but would be an excellent topic for further exploration. We have included (Abstract) information regarding the requirement for Mg^2+^ for binding to each NBS and discussion (Discussion section) regarding activation in the absence of Mg^2+^.

6) Concerning Figure 5—figure supplement 1, there are some issues with the modeling that may be due to E not being constrained to E>1 as suggested in the text (Discussion section). For example, given E=1 as reported for Figure 5—figure supplement 1B, pore opening should be independent of nucleotide binding, but that is not what is shown. Also, in Figure 5—figure supplement 1C, E=4.9e-14 implies the pore should be nearly fully closed after binding two or more nucleotides, which is clearly not what is depicted in the plot. Regardless, given the scatter in the data. we are dubious as to whether any but perhaps the most extreme model (i.e. panel e) can be ruled out, and perhaps focusing on something like this rather than arguing for a particular model would be an approach worth considering. Despite differences in RSS, model fits for, say panels a and d, are similar enough that it seems likely that they would have been considered to be not different elsewhere in the manuscript.

Thank you for pointing out this error. We have included additional binding data, acquired since the initial submission and re-fit all of the plots. For the purpose of comparison, we have also included an additional figure (Figure 5—figure supplement 2) showing the residuals of each fit. We have added further discussion of our choice of model in the Discussion section and have softened our conclusion regarding selection of the MWC model.